# GENERATIVE POINT TRACKING WITH FLOW MATCHING

## ABSTRACT

Tracking a point through a video can be a challenging task due to uncertainty arising from visual obfuscations, such as appearance changes and occlusions. Although current state-of-the-art discriminative models excel in regressing long-term point trajectory estimates—even through occlusions—they are limited to regressing to a mean (or mode) in the presence of uncertainty, and fail to capture multi-modality. To overcome this limitation, we introduce Generative Point Tracker (GenPT), a generative framework for modelling multi-modal trajectories. GenPT is trained with a novel flow matching formulation that combines the iterative refinement of discriminative trackers, a window-dependent prior for cross-window consistency, and a variance schedule tuned specifically for point coordinates. We show how our model's generative capabilities can be leveraged to improve point trajectory estimates by utilizing a best-first search strategy on generated samples during inference, guided by the model's own confidence of its predictions. Empirically, we evaluate GenPT against the current state of the art on the standard PointOdyssey, Dynamic Replica, and TAP-Vid benchmarks. Further, we introduce a TAP-Vid variant with additional occlusions to assess occluded point tracking performance and highlight our model's ability to capture multi-modality. GenPT is capable of capturing the multi-modality in point trajectories, which translates to state-of-the-art tracking accuracy on occluded points, while maintaining competitive tracking accuracy on visible points compared to extant discriminative point trackers.

## 1 INTRODUCTION

Point tracking has emerged as a popular and important approach for motion analysis in computer vision (Sand & Teller, 2008; Harley et al., 2022; Cho et al., 2024; Karaev et al., 2024b;a). Sitting between optical flow estimation (Luo et al., 2024; Saxena et al., 2024; Teed & Deng, 2020) and feature tracking (Qin et al., 2024; 2023), point tracking aims to estimate the long-range trajectories of a set of points across a video. Its utility is integral to various downstream tasks, such as structure-from-motion (He et al., 2024; Wang et al., 2021), video stabilization (Yu & Ramamoorthi, 2020; Liu et al., 2014), and 4D reconstruction (Jiang et al., 2025).

Current state-of-the-art point trackers are based on a discriminative, regression-based approach that aims to improve tracking accuracy. However, this paradigm is limited to regressing to a single mean or mode, and fails to capture the multi-modality and uncertainty inherent to point tracking, particularly during occlusions or appearance changes. This ill-posed nature of the problem motivates a generative approach capable of producing a variety of plausible solutions. To address these limitations, we introduce **GenPT**, a generative point tracker. Our primary contributions are:

1. We introduce GenPT, the **first generative point tracker trained in a modified flow matching setup** (Tong et al., 2024; Lipman et al., 2023) to capture the uncertainty inherent in point tracking.

2. We introduce **three changes to the vanilla flow matching setup** that are crucial to obtaining a competitive model: we incorporate the benefits of the iterative refinement used during training and inference of recent discriminative point trackers (Harley et al., 2022; Karaev et al., 2024a); we use a window-dependent prior to link samples across windows; and we use a specialized variance schedule in the conditional probability path that is tuned for point coordinates.

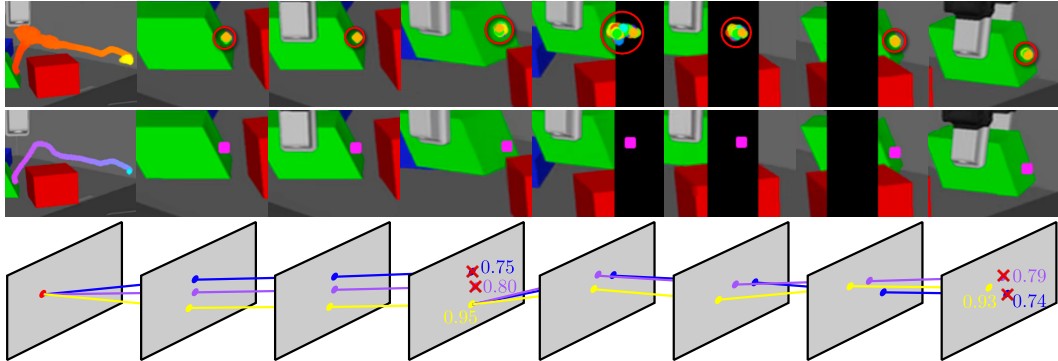

Figure 1: **Multi-modal prediction of point trajectories**. (Top) 100 randomly sampled trajectories of a single tracked point from our model, Generative Point Tracker (GenPT). The first image shows the full path of each sampled trajectory. Our model captures the multi-modality present in uncertain regions of the video. When tracking uncertainty is high, the majority of predictions are clustered around regions where the point is most likely to be. Occasionally, spurious (but valid) predictions can occur due to feature similarity elsewhere in the frame. Video is from the TAP-Vid RGB-S dataset (Doersch et al., 2022). (Middle) Ground truth for top row. (Bottom) Point trajectory estimates can be improved by utilizing a best-first search strategy on generated samples during inference, guided by the model's own confidence of its predictions in each window of observation.

3. We show how our model's **generative capabilities can be leveraged to improve point trajectory estimates by utilizing a best-first search** strategy on generated samples during inference, guided by the model's own confidence of its predictions.

We also provide architectural improvements that make our model more efficient than the state of the art, using fewer parameters and achieving faster inference, detailed in the appendix. Finally, to demonstrate the efficacy of generative tracking, we introduce a simple benchmark designed to evaluate occluded point tracking accuracy.

## 2 RELATED WORK

Point tracking is the task of estimating the motion of a set of point-wise features across a long temporal window. It lies between the short-range density of optical flow and the long-range sparsity of feature tracking. Sand & Teller (2008) introduced this problem through the particle video representation, and Harley et al. (2022) later adapted it to deep learning with their PIPs model, which leveraged RAFT's iterative refinement guided by dense feature correlation maps (Teed & Deng, 2020).

Recent models have built upon this foundation, introducing innovations like spatial attention between trajectories in CoTracker (Karaev et al., 2024b) to improve tracking through occlusions. Loco-Track (Cho et al., 2024) proposed an efficient and lightweight model alongside an improved approach to producing correlations by exchanging 2D correlation maps with 4D correlation volumes, resulting in improved tracking. CoTracker3 (Karaev et al., 2024a) improved upon CoTracker by using a simplified version of the 4D correlations from LocoTrack while simplifying other components of the CoTracker model, resulting in one of the current state-of-the-art point trackers and our main competing baseline. A key limitation shared by these models, however, is their discriminative nature; they can only regress to a single mean or mode, failing to capture the multi-modality and uncertainty inherent in point tracking. In this work, we integrate the key mechanisms that drive current state-of-the-art discriminative point trackers (*e.g.* iterative refinement guided by dense feature correlations) into a generative model, maintaining high tracking performance while enabling the modelling of multi-modality in point trajectories.

Diffusion-based models (Tong et al., 2024; Lipman et al., 2023; Luo, 2022; Song et al., 2021; Song & Ermon, 2019) have become a popular approach to generative modeling, achieving strong performance across various computer vision tasks (Esser et al., 2024; Kim et al., 2024; Karras et al., 2022; Song et al., 2021; Song & Ermon, 2019; Shafir et al., 2024; Tan et al., 2022; Liu et al., 2023; Zeng et al., 2022). Most relevant to our work are recent approaches that formulate optical flow within a diffusion-based framework (Saxena et al., 2024; Luo et al., 2024), where their ability to capture multi-modality in ill-posed cases—such as occlusions—has shown to be effective. As point tracking

faces similar challenges, this motivates a generative approach to the problem. Our work is the first to apply a generative modeling framework to point tracking. It is important to distinguish GenPT from models like ProTracker (Zhang et al., 2025) and DINTR (Nguyen et al., 2024). ProTracker produces point trajectories by chaining optical flow predictions across the video using a pre-trained discriminative optical flow model, integrating estimates in a fashion similar to a Kalman filter. DINTR applies a one-shot fine-tuning of a pre-trained image diffusion model during test time to model the temporal correspondence between two frames via interpolation, after which two-frame tracking can be performed by inspecting internal attention features. ProTracker and DINTR are not trained to maximize the likelihood of observing point trajectories, unlike GenPT, *i.e.* they can not generate samples of point trajectories from a distribution that models the underlying uncertainty. A key insight for our model (shared by Luo et al. 2024, but for optical flow estimation) is that the iterative denoising process in diffusion models closely resembles the iterative refinement used in contemporary discriminative point trackers. Our model adapts this approach to a flow matching setup (Tong et al., 2024; Lipman et al., 2023), combining the benefits of both paradigms.

## 3 METHOD

We take as input a video, $I \in \mathbb{R}^{T \times 3 \times H \times W}$, and query points, $P^q \in \mathbb{R}^{N_{Pq} \times 3}$, to be tracked in the video, where $N_{Pq}$ is the number of query points. Each query point is represented by a 3D vector describing its pixel coordinate and the video frame where it is located. Given these inputs, our goal is to predict the trajectory, $P \in \mathbb{R}^{T \times N_{Pq} \times 2}$, that tracks all query points along each frame of the video. Alongside predicted trajectories, we also want to predict visibilities, $V \in \mathbb{Z}^{T \times N_{Pq} \times 1}$, and confidences, $C \in \mathbb{R}^{T \times N_{Pq} \times 1}$. Visibility describes whether the point is occluded or not, while confidence describes the model's estimate of the error in its point trajectory estimates compared to the ground truth. Point trajectories are normalized to the range $[-1, 1]$, and visibility and confidences are bounded in the range $[0, 1]$.

### 3.1 PRELIMINARIES: FLOW-BASED GENERATIVE MODELLING

Likelihood-based generative models aim to learn the parameters, $\theta$, of a statistical model of a data distribution, $q(x)$, such that it maximizes the likelihood of observing $x \sim q(x)$ for all $x$. After training, these models can be used to generate samples, $x \sim q_\theta(x) \approx q(x)$. Diffusion-based models, like flow matching, are types of likelihood-based generative models that learn a probabilistic mapping between the data distribution, $q(x)$, and a tractable latent prior distribution (*e.g.* Gaussian), $q(z)$. The probability path between the data distribution and the prior distribution is defined by a conditional distribution that adds noise in the forward process and removes it in the reverse process.

In flow matching, both the forward and reverse processes are modeled as ordinary differential equations (ODEs) that define a deterministic conditional probability path,

$$q_l(x_l | x_L, x_1, c) = \mathcal{N}(l' x_L + (1 - l') x_1, \sigma^2), \tag{1}$$

induced by a vector field, $u_l(x_l | x_L, x_1, c) = x_L - x_1$, where $l' = (l-1)/(L-1)$, $\{l\}_{l=1}^L$, $x_1$ is a sample from the prior, $x_L$ is a sample from the data, and $c$ is optional conditioning information, such as feature maps, query points, and past predictions in our case—here we are using a form of the data-dependent coupling formulation of conditional flow matching (Liu et al., 2025; Albergo et al., 2024). The model is trained to approximate this vector field, which points toward the modes of the data distribution, by minimizing the mean squared error between the vector field and its approximation:

$$\mathcal{L}_{\text{flow-match}} = ||u_\theta(l' x_L + (1 - l') x_1 + \sigma \epsilon, l', c) - (x_L - x_1)||_2^2. \tag{2}$$

After training, samples can be generated by following this learned vector field from an initial sample from the prior through ODE integration. Unlike other diffusion-based methods, flow matching allows the prior distribution to be any parameterized distribution, not just a standard Gaussian (Liu et al., 2025; Tong et al., 2024; Albergo et al., 2024). GenPT utilizes this flexibility to incorporate initialization strategies that are inspired by those used in contemporary point tracking models. Also, instead of directly approximating the vector field, GenPT approximates the ground truth data ($x_L$) and uses an identity to produce an approximate vector field, $u_\theta(x_l, l', c) = x_\theta(x_l, l', c) - x_1$.

### 3.2 POINT TRACKING AS FLOW MATCHING

To frame point tracking as flow matching, we apply the following three crucial modifications to the vanilla flow matching setup: (i) An **iterative refinement towards the ground truth** during both

training and inference; (ii) A **window-dependent prior** to link samples across windows; (iii) A **specialized variance schedule** for the conditional probability path that is tuned for point coordinates. An ablation study on these modifications is provided in the appendix.

In our setup, point tracking involves estimating point trajectories, visibilities, and confidences in a sliding-window fashion across a video. In this section, we focus on estimating point trajectories, and we extend to visibility and confidence estimation in the next section. Given a video with $T' > T$ frames, where $T$ is our window length, we begin by splitting the video into $N_T = \lceil 2T'/T - 1 \rceil$ windows, with a window overlap of $T/2$ frames. During training, we also split the trajectory ground truths into $N_T$ overlapping windows of size $T$, resulting in $P \in \mathbb{R}^{N_T \times T \times N_{Pq} \times 2}$.

We produce two sets of predictions: estimates of the ground truth data, $\hat{P} \in \mathbb{R}^{N_T \times K \times L \times T \times N_{Pq} \times 2}$, and samples from the conditional probability path, $\tilde{P} \in \mathbb{R}^{N_T \times L \times T \times N_{Pq} \times 2}$, which are a function of the ground truth estimates. $K$ is the number of ground truth estimation refinement steps and $L - 1$ is the number of integration steps, both of which will be further explained later. For each sample along the conditional probability path, GenPT performs an iterative estimate of the ground truth data, which is used to integrate to the next sample.

Regarding notation, assume tensor broadcasting when no explicit repeats are performed, and tensors are sliced and indexed via subscript, *e.g.* $P_{n_T,1:(T/2)} \in \mathbb{R}^{(T/2) \times N_{Pq} \times 2}$ indexes the $n_T^{\text{th}}$ element of $P$ along the first axis and slices the first $T/2$ elements along the second axis.

**Window-dependent prior.** Since processing is performed in a sliding-window fashion, we aim to bootstrap estimates made in the current window with estimates made in the previous window, with the intention to improve temporal coherence of generated samples across windows. Thus, we have two cases: one where sampling is performed in the first window, and another where sampling is performed in any window past the first. We can enable this window dependency by designing priors that are conditioned on the placement of the current window, taking advantage of the flexibility offered by flow matching in choosing priors.

In the first window ($n_T = 1$), prior samples, $\tilde{P}_{n_T,1}$, are sampled from a Gaussian centred about query points, as typically done with flow matching methods:

$$\tilde{P}_{n_T,1} \sim q_1(\tilde{P}_{n_T,1}|P^q) = \mathcal{N}(P^q, \sigma_{\text{coord}}^2 \mathbf{I}) . \tag{3}$$

In subsequent windows ($n_T > 1$), prior samples are sampled piecewise along the temporal axis. Specifically, for the first $T/2$ frames in the current window, $n_T$, the prior is a Dirac delta function at the previous window's $L^{\text{th}}$ sample along its overlapping $T/2$ frames. For the last $T/2$ frames in the current window, the prior is a Gaussian centred at the previous window's $L^{\text{th}}$ sample at its final frame,

$$\tilde{P}_{n_T,1,1:(T/2)} \sim q_1(\tilde{P}_{n_T,1,1:(T/2)}|\tilde{P}_{(n_T-1),L,(T/2+1):T}) = \mathcal{N}(\tilde{P}_{(n_T-1),L,(T/2+1):T}, \mathbf{0}) , \tag{4}$$

$$\tilde{P}_{n_T,1,(T/2+1):T} \sim q_1(\tilde{P}_{n_T,1,(T/2+1):T}|\tilde{P}_{(n_T-1),L}) = \mathcal{N}(\tilde{P}_{(n_T-1),L,T}, \sigma_{\text{coord}}^2 \mathbf{I}) . \tag{5}$$

During training, we condition on $\hat{P}_{(n_T-1),K,l}$ instead of $\tilde{P}_{(n_T-1),L}$, as these should be equal after training the model.

**Conditional probability path.** During training, given prior samples, $\tilde{P}_{n_T,1}$, the ground truth, $P_{n_T}$, and $l \sim \mathcal{U}\{1, L\}$, a sample is taken from the conditional probability path at $l$,

$$\tilde{P}_{n_T,l} \sim q_l(\tilde{P}_{n_T,l}|P_{n_T}, \tilde{P}_{n_T,1}) = \mathcal{N}(l'P_{n_T} + (1 - l')\tilde{P}_{n_T,1}, l'^2 \sigma_{\text{coord}}^2 \mathbf{I}) , \tag{6}$$

where $l' = (l - 1)/(L - 1)$. This sample is a linear interpolant between the prior distribution and the data distribution and acts as the initial ground truth estimate in the current window, *i.e.* $\hat{P}_{n_T,1,l} = \tilde{P}_{n_T,l}$. In standard formulations (Tong et al., 2024; Lipman et al., 2023), the variance in the conditional probability path is set to a fixed small value or to zero, but we opt to make it a function of $l'$ that results in a linear scaling of the amount of noise in the sample. Intuitively, this means that as $l' \to 0$, the sample will be closer to the prior distribution and will contain less of the noise from the conditional probability path since the prior distribution has sufficient noise already. As $l' \to 1$, the sample will contain more of the noise in the conditional probability path since the data distribution does not have any noise. We found this to be crucial during training since it provides a balanced challenge to the model, allowing it to effectively learn how to utilize iterative estimates to reach the ground truth from any sample along the conditional probability path—see Sec. A.7 in the appendix for an ablation.

Figure 2: **Architecture overview**. GenPT is a generative point tracker based on a modified flow matching setup. **(i)** Given a query point, $P^q$, to track in a given video, GenPT starts by computing features for each frame of the video using a convnet, $F_t = \mathcal{F}_\phi(I_t)$, and sampling a neighbourhood of features around $P^q$. **(ii)** GenPT generates samples in a sliding window (*i.e.* causal) manner, where samples for each window, $\{\tilde{P}_l\}_{l=1}^L$, are composited sequentially. If in the first window, the initial sample $\tilde{P}_1$ is sampled from a Gaussian centred about $P^q$, otherwise it is initialized at sample $\tilde{P}_L$ from the previous window, with Gaussian noise added at non-overlapping frames. **(iii)** From sample $\tilde{P}_l$, a transformer $\mathcal{T}_\theta$ is used to estimate the ground truth trajectory $P$, taking $K$ refinement steps. Each step is guided by correlation features that are computed using query features and features centred about the current estimate, $\hat{P}_k$. **(iv)** The final estimate, $\hat{P}_K$, is used to integrate to the next sample, $\tilde{P}_{l+1}$. For simplicity, the figure excludes visibilities and confidences.

**Refining ground truth estimates.** Unlike vanilla flow matching, we do not compute a single estimate of the ground truth (or equivalently, the velocity field) from each sample along the conditional probability path to integrate to the next sample. As shown by discriminative point trackers (Karaev et al., 2024a; Cho et al., 2024; Harley et al., 2022) (and Luo et al. 2024 for optical flow estimation), ground truth estimates benefit from iterative refinements. We hypothesize that this is due to the limited receptive field of the feature correlations used to guide predictions. Thus, during both training and inference, after computing the initial ground truth estimate, $\hat{P}_{n_T,1,l} = \tilde{P}_{n_T,l}$, in the current window, $n_T$, at sample step $l$, GenPT refines it through $K$ residual updates: $\hat{P}_{n_T,(k+1),l} = \hat{P}_{n_T,k,l} + \Delta\hat{P}_{n_T,k,l}$.

**Sampling.** In the current window, $n_T$, and at the current sampling step $l$, the $K^{\text{th}}$ ground truth estimate, $\hat{P}_{n_T,K,l}$, is used to compute the vector field, $\tilde{P}^{\text{flow}}$, which is used to integrate to the next sample, $\tilde{P}_{n_T,(l+1)}$, through Euler integration, $\tilde{P}_{n_T,(l+1)} = \tilde{P}_{n_T,l} + \Delta l' \tilde{P}^{\text{flow}}$, where $\tilde{P}^{\text{flow}} = \hat{P}_{n_T,K,l} - \tilde{P}_{n_T,1}$, and $\tilde{P}_{n_T,1}$ is the prior sample.

**Point trajectory training loss.** Point trajectory ground truth estimates are supervised by the following loss:

$$\mathcal{L}_{\text{coord}} = \frac{0.05}{N_T K} \sum_{n_T=1}^{N_T} \sum_{k=1}^{K-1} |P_{n_T} - \hat{P}_{n_T,(k+1),l}| . \tag{7}$$

We opt for an L1 loss instead of the mean squared error, as motivated by Saxena et al. (2024). Visibility and confidence estimation are modeled in much the same way as point trajectory estimation. The next section provides a detailed description of the full approach.

## 3.3 MODEL ARCHITECTURE

GenPT is a transformer-based (Vaswani et al., 2017) model that uses factorized temporal and spatial attention blocks, as introduced in CoTracker (Karaev et al., 2024b). We make architectural improvements to the CoTracker3 (Karaev et al., 2024a) transformer to arrive at a more efficient model, using 12M parameters instead of 25M and operating at twice the inference speed, details of which can be found in Secs. A.2 and A.16 of the appendix, respectively. In addition to point trajectory ground truth estimates and samples, GenPT produces visibility and confidence ground truth estimates, $\hat{V} \in \mathbb{R}^{N_T \times K \times L \times T \times N_{P^q} \times 1}$ and $\hat{C} \in \mathbb{R}^{N_T \times K \times L \times T \times N_{P^q} \times 1}$, respectively, and visibility and confidence samples, $\tilde{V} \in \mathbb{R}^{N_T \times L \times T \times N_{P^q} \times 1}$ and $\tilde{C} \in \mathbb{R}^{N_T \times L \times T \times N_{P^q} \times 1}$, respectively. There are four stages of processing: (i) Extraction of visual features; (ii) Initialization of samples and ground truth estimates; (iii) Iterative updates of ground truth estimates; (iv) Using ground truth estimates to

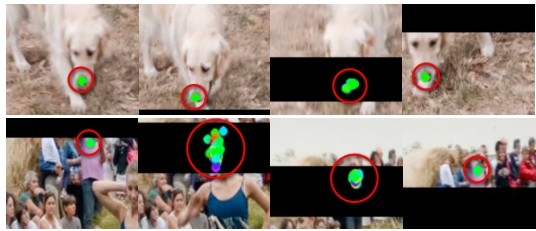

Figure 3: **Multi-modal prediction due to spatial homogeneity (left) and self-occlusion (right).** (Left) When the tracked point is a feature that is spatially homogeneous, predictions become increasingly varied as time moves on, especially in the presence of occlusion (top-left). As expected, these predictions cluster around neighbouring visually similar areas. Videos shown are from the TAP-Vid RGB-S dataset (Doersch et al., 2022). (Right) As the subject rotates, the tracked point becomes increasingly occluded, resulting in increasingly varied predictions that cluster around the expected location of the point. Videos shown are from the TAP-Vid DAVIS dataset (Doersch et al., 2022).

integrate to the next sample. Details of the training and sampling loop are provided in Algs. 1 and 2 in the appendix. An overview of our architecture is provided in Fig. 2.

**Extracting features.** Point tracking begins with computing features $F_t = \mathcal{F}_\phi(I_t)$ for each frame of the video, where $\mathcal{F}_\phi$ is a residual convnet (He et al., 2016) that matches the encoder architecture used in RAFT (Teed & Deng, 2020) with parameters $\phi$. After the features are computed, a multi-scale pyramid of the features is computed at $S = 3$ scales, resulting in a set of feature maps $\{F^s \in \mathbb{R}^{T' \times D_\mathcal{F} \times H/2^{s+1} \times W/2^{s+1}}\}_{s=1}^S$, where $D_\mathcal{F} = 128$ is the number of feature channels. Query points are replicated $T$ times, resulting in $P^q \in \mathbb{R}^{T \times N_{P^q} \times 3}$. At the $7 \times 7$ neighbourhood centred about each query point, $\Omega(P^q)$, features are bilinearly sampled at each scale, $f^{\Omega(P^q)} \in \mathbb{R}^{S \times T \times N_{P^q} \times D_\mathcal{F} \times 49}$. After sampling query neighbourhood features, the set of feature

Figure 4: **Target reacquisition post-occlusion.** When the point being tracked is uniquely identifiable in the scene, predictions made behind an occluder cluster around the expected location of the point. However, when the occluder passes, the variance of predictions rapidly shrinks. Videos shown are from the TAP-Vid DAVIS dataset (Doersch et al., 2022).

maps is sliced into $N_T$ overlapping windows of size $T$ in preparation for the refinement stage, resulting in $\{F^s \in \mathbb{R}^{N_T \times T \times D_\mathcal{F} \times H/2^{s+1} \times W/2^{s+1}}\}_{s=1}^S$.

**Initialization.** After extracting query features, GenPT initializes samples for the current window, $n_T$. During training, samples are initialized from the conditional probability path at a randomly sampled step $l \sim \mathcal{U}\{1, L\}$, as described in Sec. 3.2. For visibilities and confidences, the conditional probability paths are:

$$\tilde{V}_{n_T,l} \sim q_l(\tilde{V}_{n_T,l}|V_{n_T}, \tilde{V}_{n_T,1}) = \mathcal{N}(l'V_{n_T} + (1 - l')\tilde{V}_{n_T,1}, l'^2\sigma_{\text{vis}}^2\mathbf{I}) , \quad (8)$$

$$\tilde{C}_{n_T,l} \sim q_l(\tilde{C}_{n_T,l}|C_{n_T}, \tilde{C}_{n_T,1}) = \mathcal{N}(l'C_{n_T} + (1 - l')\tilde{C}_{n_T,1}, l'^2\sigma_{\text{conf}}^2\mathbf{I}) . \quad (9)$$

During inference, samples are initialized from the window-dependent prior (*i.e.* $l = 1$), which follows the same form as for point trajectories as described in Sec. 3.2, except in the first window ($n_T = 1$), where the prior is as follows:

$$\tilde{V}_{n_T,1} \sim q_1(\tilde{V}_{n_T,1}) = \mathcal{N}(0, \sigma_{\text{vis}}^2\mathbf{I}) , \quad (10)$$

$$\tilde{C}_{n_T,1} \sim q_1(\tilde{C}_{n_T,1}) = \mathcal{N}(0, \sigma_{\text{conf}}^2\mathbf{I}) . \quad (11)$$

Initialization of ground truth estimates of visibilities and confidences follows the same approach as for point trajectories as described in Sec. 3.2.

**Refining ground truth estimates.** As mentioned previously, the sample at step $l$ in the current window, $n_T$, is used as the initial ground truth estimate: $\{\hat{P}_{n_T,1,l}, \hat{V}_{n_T,1,l}, \hat{C}_{n_T,1,l}\} = \{\tilde{P}_{n_T,l}, \tilde{V}_{n_T,l}, \tilde{C}_{n_T,l}\}$. From here, a transformer, $\mathcal{T}_\theta$, is used to refine the ground truth estimate

Table 1: **Performance on TAP-Vid, PointOdyssey, and Dynamic Replica benchmarks**. Also shown are results when taking the best of N sampled trajectories for each sliding window, using either greedy search guided by the model's predicted confidence or an oracle (*i.e.* their distance to the ground truth). The oracle results highlights our model's performance ceiling. TAP-Vid results are measured in $\delta_{\text{avg}}^{\text{vis}}$ ($\uparrow$). † Using pre-trained model weights.

| Method | Train | Kin. | RGB. | DAV. | Robo. | Avg. | Method | Train | PointOdy. $\delta_{\text{avg}}^{\text{vis}}$ | $\delta_{\text{avg}}^{\text{occ}}$ | Dyn. Rep. $\delta_{\text{avg}}^{\text{vis}}$ | $\delta_{\text{avg}}^{\text{occ}}$ |
|---|---|---|---|---|---|---|---|---|---|---|---|---|
| CoTracker2.1† | Kub | 61.5 | 77.2 | 76.1 | OOM | 71.6 | CoTracker2.1† | Kub | 60.9 | 49.4 | 81.2 | 53.0 |
| CoTracker3† | Kub | 65.5 | 78.8 | 76.7 | 75.0 | 74.0 | CoTracker3† | Kub | 68.2 | 55.0 | 84.7 | 52.9 |
| CoTracker3† | Kub+15k | 68.5 | 82.7 | 77.4 | **80.4** | 77.3 | CoTracker3† | Kub+15k | 69.3 | 55.2 | 83.6 | 48.4 |
| LocoTrack† | Kub | 67.8 | **83.3** | 75.5 | 77.8 | 76.1 | LocoTrack† | Kub | 64.0 | 48.7 | 83.0 | 34.6 |
| CoTracker3 | Kub | 64.6 | 78.0 | 74.5 | 75.1 | 73.0 | CoTracker3 | Kub | 61.4 | 47.8 | 77.7 | 40.0 |
| CoTracker3 | PO | 67.2 | 79.3 | 77.4 | 75.5 | 74.8 | CoTracker3 | PO | 69.1 | 54.7 | 83.1 | 45.3 |
| GenPT (ours) | Kub | 65.9 | 82.2 | 76.3 | 77.7 | 75.5 | GenPT (ours) | Kub | 66.0 | 53.1 | 83.2 | 48.8 |
| GenPT (ours) | PO | 67.7 | 79.4 | 77.7 | 79.0 | 75.9 | GenPT (ours) | PO | 70.2 | 57.9 | 85.4 | 53.8 |
| ↳greedy, best of 5 | | 68.6 | 81.7 | **78.3** | 80.2 | 77.2 | ↳greedy, best of 5 | | 71.4 | **58.9** | **85.7** | **54.2** |
| ↳greedy, best of 10 | | **68.7** | 82.2 | **78.3** | **80.4** | **77.4** | ↳greedy, best of 10 | | **71.6** | 58.8 | 85.6 | **54.2** |
| ↳oracle, best of 5 | | 70.5 | 84.3 | 80.1 | 82.2 | 79.3 | ↳oracle, best of 5 | | 74.0 | 62.7 | 87.3 | 57.7 |
| ↳oracle, best of 10 | | **71.7** | **85.8** | **80.5** | **83.0** | **80.2** | ↳oracle, best of 10 | | **75.1** | **64.2** | **88.0** | **59.2** |

through $K$ residual updates, as described in Sec. 3.2. To facilitate updates, GenPT uses 4D correlation features (Karaev et al., 2024a) computed using query neighbourhood features, $f^{\Omega(P^q)}$, and feature maps, $F^s$, at each step $k$ of $K$, cropped about each point in the trajectory, $\hat{P}_{n_T,k,l}$. These correlation features help guide GenPT's point estimates towards locations where the query point is most likely to be located. In addition to correlation features and the current ground truth estimate, $\{\hat{P}_{n_T,k,l}, \hat{V}_{n_T,k,l}, \hat{C}_{n_T,k,l}\}$, the transformer is given the normalized sample step $l'$, a temporal positional encoding, and a flag indicating whether it is in the first window or any subsequent window.

**Sampling.** Point trajectories are sampled according to the procedure described in Sec. 3.2. Visibilities and confidences are sampled in a similar fashion, where refined ground truth estimates are used to integrate to the next sample.

**Training loss.** We train our model with the following loss, $\mathcal{L} = \mathcal{L}_{\text{coord}} + \mathcal{L}_{\text{vis}} + \mathcal{L}_{\text{conf}}$, where,

$$\mathcal{L}_{\text{vis}} = \frac{1}{N_T K} \sum_{n_T=1}^{N_T} \sum_{k=1}^{K-1} \texttt{BCE}(V_{n_T}, \texttt{sigmoid}(\hat{V}_{n_T,(k+1),l})), \tag{12}$$

$$\mathcal{L}_{\text{conf}} = \frac{1}{N_T K} \sum_{n_T=1}^{N_T} \sum_{k=1}^{K-1} |C_{n_T,(k+1),l} - \hat{C}_{n_T,(k+1),l}|. \tag{13}$$

`BCE` is Binary Cross Entropy and $C_{n_T,(k+1),l} = 1 - \min((\texttt{stopgrad}(\hat{P}_{n_T,(k+1),l}) - P_{n_T})^2, 16^2)/16^2$. We take a unique approach in supervising confidence predictions. Instead of treating confidence as a classification of whether the predicted point is within a certain distance threshold of the ground truth, we treat it as a regression of the model's prediction error. We find that this slightly improves tracking accuracy—an ablation is provided in Sec. A.11 in the appendix.

## 4 EXPERIMENTS

In this section, we evaluate the efficacy of our model in tracking points and capturing their uncertainty. We start by outlining our training details and evaluation setup, followed by an evaluation on standard benchmarks and our occlusion-focused variant. Then we provide analyses on experiments that focus on the multi-modal aspect of our model, a unique aspect not captured by other models.

**Training details.** To train GenPT, we use the PointOdyssey dataset (Zheng et al., 2023) as the source of ground truth point trajectories and visibilities. PointOdyssey is a synthetic dataset for the training and evaluation of point tracking algorithms. It contains a wide variety of motion profiles, materials, lighting, 3D assets, and atmospheric effects. We also include a GenPT model trained on a version of the Kubric dataset (Doersch et al., 2022) provided by the CoTracker3 authors. Kubric is a synthetic dataset with simpler imagery and dynamics compared to PointOdyssey.

**Evaluation setup.** We evaluate our model on three standard benchmarks: PointOdyssey (test split), Dynamic Replica (Karaev et al., 2023) (validation split), and TAP-Vid (Doersch et al., 2022). TAP-

Vid contains four datasets: Kinetics, RGB-S, DAVIS, and Kubric. However, we replace Kubric with the RoboTAP test set (Vecerik et al., 2024) as Kubric does not contain a test split. TAP-Vid does not include ground truth for occluded points, so to further explore the capability of our model to capture uncertainty under occlusions, we introduce a variant of TAP-Vid by translating a 100-pixel wide black bar through each video left-to-right, right-to-left, bottom-to-top, and top-to-bottom (examples seen in Fig. 3 and 4), where the bar starts translating at the first frame and ends at the last frame.

We compare our model (12M parameters) against four pre-trained models: CoTracker2.1 (an improved, yet unpublished version of CoTracker by the same authors with publicly available code[1], with 45M parameters), CoTracker3 trained on Kubric (25M parameters) (Karaev et al., 2024a), CoTracker3 trained on Kubric and 15K real-world videos via self-supervision (25M parameters) (Karaev et al., 2024a), and LocoTrack-B trained on Kubric (12M parameters) (Cho et al., 2024).

GenPT is trained on PointOdyssey, while many prior works use Kubric. Since Kubric is generated with a script that introduces random variations such as random cropping, the exact datasets differ slightly across works. The CoTracker3 authors have released their Kubric variant, so we use this to train a comparable version of GenPT. We also train two new CoTracker3 models for additional comparison: one on PointOdyssey and one on Kubric, both using the authors' public code and a training configuration that is identical to GenPT's (batch size, number of training steps, $N_{Pq}$, window size, video length, and data augmentations). These CoTracker3 models serve as our main baselines. We use the following standard metrics to evaluate tracking accuracy: $\delta_{\mathrm{avg}}^{\mathrm{vis}}$ ($\uparrow$) for visible points and $\delta_{\mathrm{avg}}^{\mathrm{occ}}$ ($\uparrow$) for occluded points.

### 4.1 STANDARD BENCHMARKS

Table 1 contains quantitative results on the TAP-Vid, PointOdyssey, and Dynamic Replica benchmarks. In a single shot—without leveraging GenPT's capability to sample and pick the best of N trajectories—and averaging across all standard benchmarks and all models, the pre-trained CoTracker3 model (trained on Kubric + 15K) performs the best on $\delta_{\mathrm{avg}}^{\mathrm{vis}}$, while our PointOdyssey-trained model performs the best on $\delta_{\mathrm{avg}}^{\mathrm{occ}}$. When comparing our model with CoTracker3 when both are identically trained on PointOdyssey, we can see that our model outperforms CoTracker3 on all test sets, with an average $1.5\%$ gap on $\delta_{\mathrm{avg}}^{\mathrm{vis}}$ and a $5.9\%$ gap on $\delta_{\mathrm{avg}}^{\mathrm{occ}}$. When both are identically trained on Kubric, the average gap is $4.2\%$ on $\delta_{\mathrm{avg}}^{\mathrm{vis}}$ and $7.1\%$ on $\delta_{\mathrm{avg}}^{\mathrm{occ}}$. Recall that our model contains less than half the number of parameters as CoTracker3. Also, our Kubric-trained model manages to outperform the Kubric-pre-trained CoTracker3 on TAP-Vid by $1.5\%$, despite being trained on less data.

### 4.2 TAP-VID SLIDING OCCLUDER BENCHMARK

Table 2 contains quantitative results on our TAP-Vid sliding occluder benchmark. Our model maintains a similar lead on $\delta_{\mathrm{avg}}^{\mathrm{vis}}$ as it did before on TAP-Vid when comparing against the CoTracker3 model we trained. Our PointOdyssey-trained model outperforms all models on $\delta_{\mathrm{avg}}^{\mathrm{occ}}$ by a significant margin, managing a $9.5\%$ lead over our PointOdyssey-trained CoTracker3, a $10.6\%$ lead over CoTracker3 (Kub + 15k), a $10.8\%$ lead over CoTracker3 (Kub), a $14.1\%$ lead over CoTracker2.1, and a $25.4\%$ lead over LocoTrack. Our Kubric-trained model maintains a similar $\delta_{\mathrm{avg}}^{\mathrm{occ}}$ gap with most models, except there is a larger $12.2\%$ lead when compared with our identically-trained CoTracker3.

### 4.3 QUALITATIVE RESULTS: CAPTURING MULTI-MODALITY

Here we show our model's ability to sample many plausible trajectories in the presence of occlusion or other visual obfuscations. In Fig. 1, 3, and 4 we can see examples where predictions of the tracked point grow increasingly varied as it is occluded, with most predictions clustering around the region where the point would most likely be (*i.e.* a mode), and occasional spurious predictions occurring elsewhere due to feature similarity, which is expected as it is still a plausible mode. In Fig. 3 (left), we show that when the tracked point is a feature that is spatially homogeneous (*e.g.* a point on a texture-less surface), predictions become increasingly varied as time moves on, resulting in predictions clustered around neighbouring visually similar areas. In Fig. 3 (right), we can see additional examples of occlusion causing an increased variance in predicted point locations, with predictions clustering around the expected location of the point. In Fig. 4, we can see that when the point being tracked is uniquely identifiable in the scene, then the variance of predictions will remain

---

[1] https://github.com/facebookresearch/co-tracker

Table 2: **Performance on TAP-Vid sliding occluder benchmark**. A 100-pixel wide black bar is translated across each video in various directions. Results are averaged across directions. Also shown are results when taking the best of N sampled trajectories for each sliding window, using either greedy search guided by the model's predicted confidence or an oracle (*i.e.* their distance to the ground truth). The oracle results highlights our model's performance ceiling. † Using pre-trained model weights.

| Method | Train | Kinetics $\delta_{avg}^{vis}$ | $\delta_{avg}^{occ}$ | RGB-S $\delta_{avg}^{vis}$ | $\delta_{avg}^{occ}$ | DAVIS $\delta_{avg}^{vis}$ | $\delta_{avg}^{occ}$ | RoboTAP $\delta_{avg}^{vis}$ | $\delta_{avg}^{occ}$ | Avg. $\delta_{avg}^{vis}$ | $\delta_{avg}^{occ}$ |
|---|---|---|---|---|---|---|---|---|---|---|---|
| CoTracker2.1† | Kub | 59.1 | 44.0 | 63.3 | 45.6 | 74.2 | 59.1 | OOM | OOM | 65.5 | 49.6 |
| CoTracker3† | Kub | 64.1 | 47.9 | 69.4 | 42.5 | 76.1 | 61.2 | 72.8 | 60.2 | 70.6 | 52.9 |
| CoTracker3† | Kub+15k | 67.7 | 50.0 | 75.1 | 37.4 | 76.7 | 62.0 | 78.8 | 63.0 | 74.6 | 53.1 |
| LocoTrack† | Kub | 67.5 | 34.8 | 76.7 | 38.2 | 75.0 | 40.0 | 76.6 | 40.1 | 74.0 | 38.3 |
| CoTracker3 | Kub | 63.2 | 44.9 | 70.7 | 38.5 | 73.6 | 56.6 | 72.4 | 56.5 | 70.0 | 49.1 |
| CoTracker3 | PO | 66.5 | 49.3 | 74.0 | 46.3 | 77.2 | 62.8 | 74.2 | 58.4 | 73.0 | 54.2 |
| GenPT (ours) | Kub | 65.4 | 52.9 | 76.4 | 62.9 | 75.5 | 63.5 | 76.4 | 66.0 | 73.4 | 61.3 |
| GenPT (ours) | PO | 67.2 | 54.7 | 76.1 | 66.4 | 77.6 | 66.5 | 77.9 | 67.4 | 74.7 | 63.7 |
| ↳greedy, best of 5 | | 68.1 | **56.0** | 78.7 | 69.4 | 77.7 | 66.9 | **79.1** | 69.1 | 75.9 | 65.3 |
| ↳greedy, best of 10 | | **68.2** | 55.9 | **79.1** | 69.8 | 77.9 | 67.6 | **79.1** | 69.2 | 76.1 | 65.6 |
| ↳oracle, best of 5 | | 70.4 | 60.0 | 82.4 | 74.4 | 80.0 | 70.3 | 81.6 | 73.7 | 78.6 | 69.6 |
| ↳oracle, best of 10 | | **71.5** | **61.8** | **84.3** | **76.9** | **80.5** | **71.1** | **82.6** | **75.6** | **79.7** | **71.3** |

small, only temporarily increasing under the presence of occlusion. Additional qualitative results can be found in the form of videos provided in the supplementary material.

## 4.4 BEST OF N SAMPLES

Here we take inspiration from other computer vision applications of generative modelling (Kolotouros et al., 2021; Jayaraman et al., 2019) and consider an evaluation that takes into account the stochastic nature of our problem set, with the intention of highlighting how our method explores the potential multiple modes in the solution space, especially in the presence of uncertainty. We create two categories of experiments under our benchmarks: an oracle experiment where we construct predictions by taking the best of N sampled trajectories for each window, using the ground truth as reference for picking the best prediction; and a variant where we use the model's confidence as guidance for picking the best prediction. Tables 1 and 2 shows our results with oracle guidance (labelled "oracle") and confidence guidance (labelled "greedy") using our PointOdyssey-trained model.

With the oracle as guidance and taking five samples, we see a 3.3% $\delta_{avg}^{vis}$ and 4.9% $\delta_{avg}^{occ}$ improvement averaged across all benchmarks. We see further improvements when taking ten samples, with a 4.2% $\delta_{avg}^{vis}$ and 6.4% $\delta_{avg}^{occ}$ improvement. These results represent our model's performance ceiling. With confidence predictions as guidance and taking five samples, we see a 1% $\delta_{avg}^{vis}$ and 1% $\delta_{avg}^{occ}$ improvement. With ten samples, there is a 1.1% $\delta_{avg}^{vis}$ and 1.1% $\delta_{avg}^{occ}$ improvement. Using the model's own confidence allows us to further increase the gap between our model and the CoTracker3 model we trained, even outperforming the pre-trained CoTracker3 model (Kub. + 15k) on $\delta_{avg}^{vis}$ by 1.4% when taking five samples. Although there is consistent improvement when increasing the number of samples, there is a gap between using the model's confidence as guidance and using the oracle. This suggests that both our search strategy and confidence estimation can be improved. However, we experimented with using beam search (Sec. A.12 in the appendix) and found no favourable gains over greedy search, leading us to believe that most gains will come from improving confidence accuracy.

## 5 DISCUSSION

We have introduced GenPT, a novel generative point tracker that addresses the limitations of conventional discriminative models by modelling the multi-modality inherent to point tracking. By adapting a flow matching setup with iterative refinement, a window-dependent prior, and a specialized variance schedule, GenPT effectively captures uncertainty, particularly behind occlusions. Our extensive evaluations, including a new occluder benchmark, demonstrate GenPT's superior ability to track occluded points and explore plausible modes in the solution space. Furthermore, we showed how a best-first search strategy on generated samples, guided by the model's confidence, can be used to improve tracking accuracy. GenPT achieves its state-of-the-art performance while using significantly fewer parameters and operating at a faster inference speed compared to baselines.

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

# A  APPENDIX

## A.1  BROADER IMPACT.

Point tracking is a foundational computer vision task that can be used to bootstrap models for a wide range of higher-level applications, including navigation for autonomous vehicles and video surveillance. The improvements introduced by GenPT, particularly its ability to model multi-modal trajectories, could enhance these applications. As with any powerful technology, its application can have both positive and negative societal effects, and it is crucial to consider the context of its use.

## A.2  TRANSFORMER ARCHITECTURE DETAILS

GenPT uses a transformer, $\mathcal{T}_\theta$, that processes point trajectory, visibility, and confidence inputs as spatiotemporal tokens, using factorized temporal and spatial attention blocks, similarly to Co-Tracker (Karaev et al., 2024b). Fig. 5 shows a detailed overview of the transformer architecture.

As input, the transformer takes in the forward and backward per-frame coordinate displacements computed from $\hat{P}_{n_T,k,l}$, visibilities, confidences, the current noise level, a flag that indicates which prior the sample was initialized from, and frame indices (see Alg. 1 and 2 for details). It uses correlations features through a separate conditioning pipeline, explained later. The transformer outputs a refinement to ground truth estimates of point trajectories, visibilities, and confidences.

To improve the parameter efficiency of GenPT, we make various modifications to the base CoTracker architecture that it's based off of. We reduce the number of spatial cross-attention blocks from six to one. We eliminate the spatial self-attention that occurs between virtual points. We reduce the number of feature pyramid levels from four to three. We reduce the attention MLP multiplier from four to two. Taking inspiration from the Hourglass Diffusion model (Crowson et al., 2024), we eliminate bias in all linear layers and condition using correlation features that are fed to Adaptive RMS Norm layers (Zhang & Sennrich, 2019; Huang & Belongie, 2017) placed within the temporal self-attention blocks instead of concatenating the correlation features to the input; RMS Norm layers are used elsewhere in place of Layer Norm; and the attention MLP uses a LinearGEGLU layer (Crowson et al., 2024) instead of a linear layer followed by a GELU activation. Finally, we initialize the weights of the last linear layer of the transformer, the last linear layer in the attention MLPs, the linear layer applied after the attention operation (not shown in Fig. 5), and the linear layer in the Adaptive RMS Norm layers to values close to zero. The weight initialization ensures that the first training step will result in a near-zero update, which helps with early training stability. Our architectural changes results in a 12M parameter model *vs*. CoTracker3's 25M parameters.

## A.3  MORE TRAINING DETAILS

To train our model to process videos in an online fashion, we follow the unrolled window training scheme of CoTracker3. During training, we randomly sample a $384 \times 512$ video with $T' > T$ frames, where $T' = 24$ is the video length, $T = 16$ is the window length, and the sliding window stride is $T/2$, resulting in $N_T = 2$ windows. Alongside each video we randomly sample 128 ground truth trajectories ($N_{P^q} = 128$). The model produces ground truth estimates for each window in a sequential fashion, starting from a sample from the conditional probability path that is sampled at noise level $l \sim \mathcal{U}\{1, L\}$, where $L = 1000$ (or in a continuous fashion, one can directly sample $l' \sim \mathcal{U}(0, 1)$). Each window consists of $K = 4$ ground truth estimates, with an estimate being a refinement to the previous estimate. We train on four NVIDIA L40S 48GB GPUs for 400K iterations, with a batch size of one per GPU (*i.e.* a total batch size of four). We train using the AdamW optimizer (Loshchilov & Hutter, 2019) with weight decay set to 0.001, $\beta_1$ set to 0.9, $\beta_2$ set to 0.999, and a max learning rate of $5e - 4$, using the "1cycle" learning rate scheduler with linear annealing (Smith & Topin, 2019). Gradients norms are clipped to one.

As mentioned in the main manuscript, we evaluated against pre-trained CoTracker and LocoTrack models, and against CoTracker3 models that we trained from scratch. We tried training GenPT with the exact same data and data pre-processing pipeline used for CoTracker3 pre-trained on Kubric (with $T' = 64$ and $N_{P^q} = 384$), but we quickly ran out of memory. To keep comparisons fair against CoTracker3, we instead trained CoTracker3 from scratch with the settings we described above. In addition to all this, it is worth mentioning that CoTracker2.1 and LocoTrack were trained on different

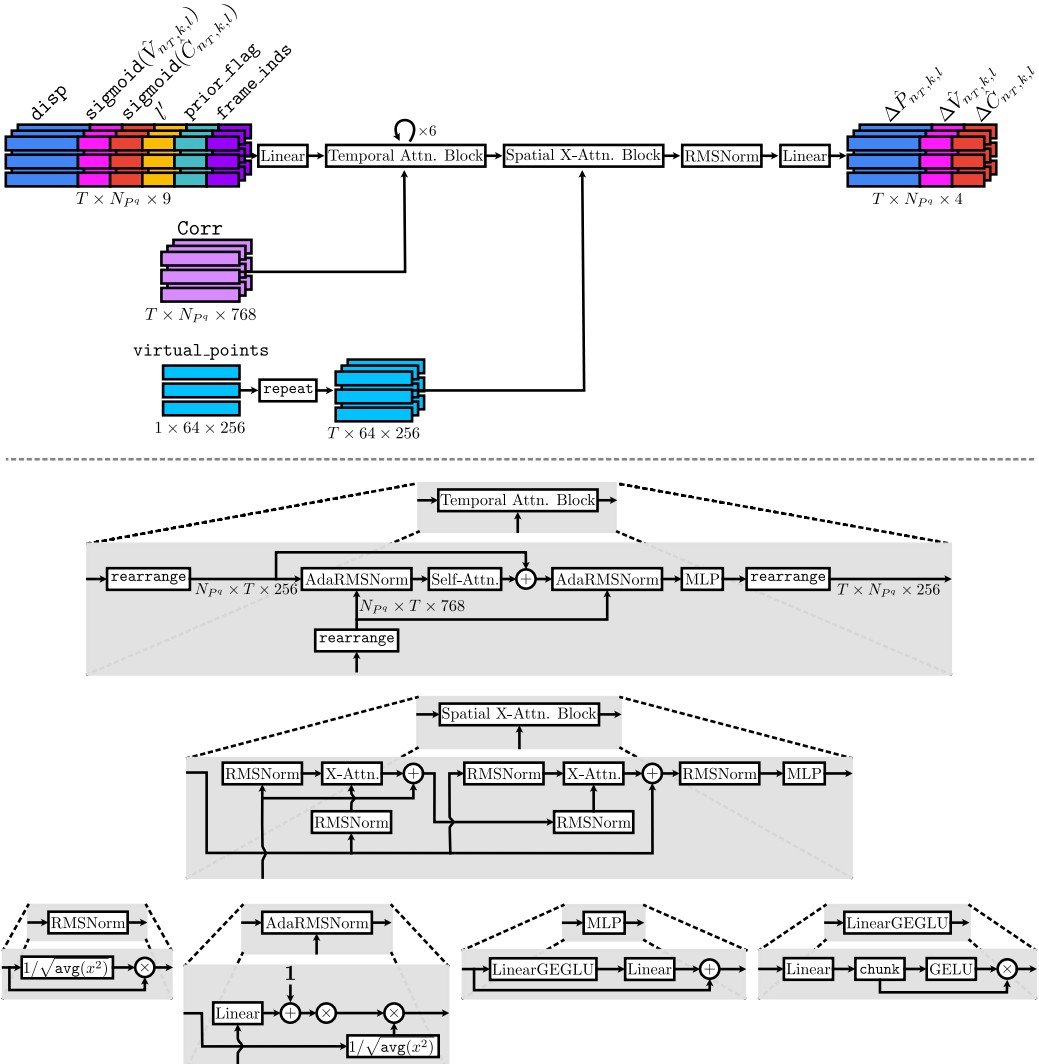

Figure 5: **Transformer architecture overview**. GenPT uses a transformer, $\mathcal{T}_\theta$, that processes point trajectory, visibility, and confidence inputs as spatiotemporal tokens, using factorized temporal and spatial attention blocks, similarly to CoTracker (Karaev et al., 2024b). `disp` contains the forward and backward per-frame coordinate displacement computed from $\hat{P}_{n_T,k,l}$.

instantiations of Kubric from each other and CoTracker3. Specifically, Kubric is a dataset that is generated through a script, which contains random augmentations such as random cropping. When CoTracker2.1 and LocoTrack were developed, they were trained on their own generated version of Kubric that is unreleased, making it impossible to provide an apples-to-apples comparison on the basis of architecture alone. Fortunately, with CoTracker3, the authors released the exact Kubric dataset they generated and used, which allowed us to provide a fair comparison when both GenPT and CoTracker3 are trained on Kubric. We opted against training CoTracker2.1 and LocoTrack from scratch on PointOdyssey and Kubric since the CoTracker3 authors already showed its superiority over these models.

## A.4 MORE EVALUATION DETAILS

Kinetics consists of 1144 real-world videos from the Kinetics-700-2020 validation set (Carreira & Zisserman, 2017), mostly containing scenes of complex motion and appearance of people and objects. RGB-S is a synthetic dataset consisting of 50 videos recorded from a simulated robotic stacking

environment containing texture-less objects that are challenging to track (see Fig. 3). DAVIS is a dataset containing 30 real-world videos from the DAVIS 2017 validation set (Perazzi et al., 2016) with a wide variety of motion and appearance across various people and objects. RoboTAP is a dataset containing 265 real-world robotics manipulation videos.

PointOdyssey's test split contains 12 videos of synthetic scenes with a wide variety of motion profiles, materials, lighting, 3D assets, and atmospheric effects. Dynamic Replica's validation split contains 20 videos of synthetic scenes that are visually similar to scenes in PointOdyssey. Both PointOdyssey and Dynamic Replica contain ground truth annotations for occluded points. Due to the length of PointOdyssey and Dynamic Replica videos and abundance of ground truth trajectories, we only evaluate 256 trajectories per video and truncate each video's length to 300 frames.

For each dataset, we sample query points from the first frame and remove trajectories whose first point is not visible, as was done in PIPs (Harley et al., 2022) and PIPs++ (Zheng et al., 2023). Each video is resized to $384 \times 512$.

We enable joint tracking across trajectories for our model and all CoTracker models, but we disable any use of support points (Karaev et al., 2024b) to minimize unnecessary computations. For evaluation, we set the number of update steps $K = 6$ for all competing models. For GenPT, we set $K = 3$ and $L = 3$ so that the total number of function evaluations per window is 6 (recall $L$ is the number of noise levels and $L - 1$ is the number of integration steps).

## A.5 EFFECT OF ITERATIVE REFINEMENT DURING TRAINING

In this experiment, we ablate iterative refinement during training and study its effect on tracking accuracy. Specifically, we train GenPT with $K = 1$ and compare it with GenPT trained with $K = 4$. Both GenPT models are trained on PointOdyssey and in a shortened training setup, where we train for 100K steps instead of 400K and use a total batch size of two instead of four.

Table 3 shows our quantitative results, which shows that iterative refinement provides a substantial boost in tracking performance for GenPT. We hypothesize that the reason it may be so important is because the transformer is given cropped correlation features that have a limited receptive field, and so it needs to take multiple steps in order to move its receptive field to an area that most likely contains the ground truth. Additionally, we believe it serves as a form of data augmentation, where it is being trained to correct its own errors.

Table 3: **Effect of iterative refinement during training on the TAP-Vid (top-left), PointOdyssey (top-right, left), Dynamic Replica (top-right, right), and TAP-Vid sliding occluder (bottom) benchmarks**. Each model is trained in a shortened training setup. TAP-Vid results are measured in $\delta_{\text{avg}}^{\text{vis}}$ ($\uparrow$).

| Iterative refine | Kin. | RGB. | DAV. | Robo. | Avg. | Iterative refine | PointOdy. $\delta_{\text{avg}}^{\text{vis}}$ | $\delta_{\text{avg}}^{\text{occ}}$ | Dyn. Rep. $\delta_{\text{avg}}^{\text{vis}}$ | $\delta_{\text{avg}}^{\text{occ}}$ |
|---|---|---|---|---|---|---|---|---|---|---|
| Disabled | 64.0 | 76.9 | 64.7 | 75.1 | 70.2 | Disabled | 58.4 | 46.5 | 76.5 | 42.9 |
| Enabled | **66.5** | **77.9** | **74.2** | **78.0** | **74.1** | Enabled | **64.5** | **51.6** | **81.9** | **48.1** |

| Iterative refine | Kinetics $\delta_{\text{avg}}^{\text{vis}}$ | $\delta_{\text{avg}}^{\text{occ}}$ | RGB-S $\delta_{\text{avg}}^{\text{vis}}$ | $\delta_{\text{avg}}^{\text{occ}}$ | DAVIS $\delta_{\text{avg}}^{\text{vis}}$ | $\delta_{\text{avg}}^{\text{occ}}$ | RoboTAP $\delta_{\text{avg}}^{\text{vis}}$ | $\delta_{\text{avg}}^{\text{occ}}$ | Avg. $\delta_{\text{avg}}^{\text{vis}}$ | $\delta_{\text{avg}}^{\text{occ}}$ |
|---|---|---|---|---|---|---|---|---|---|---|
| Disabled | 63.5 | 43.5 | 72.7 | 53.2 | 64.0 | 43.4 | 73.4 | 51.6 | 68.4 | 47.9 |
| Enabled | **65.9** | **50.9** | **73.7** | **59.2** | **73.6** | **59.0** | **76.4** | **61.9** | **72.4** | **57.8** |

## A.6 WINDOW-DEPENDENT PRIOR *vs*. WINDOW-INDEPENDENT PRIOR

In this experiment, we ablate the window-dependent prior of GenPT and study its effect on tracking accuracy. We train two GenPT models in a shortened training setup on PointOdyssey, as described previously. The first is our standard model, which has two priors: one for the first window, and another for subsequent windows, which is designed to enable window linking. The second model is a variant that uses a single prior for all windows, which is set to the prior used when initializing samples in the first window in the standard model.

Table 4 shows our quantitative results, which shows that the window-dependent prior provides a substantial boost in tracking performance. In other words, past the first window, it is beneficial to initialize samples centred around the samples generated in the previous window.

Table 4: **Window-dependent prior *vs*. window-independent prior on the TAP-Vid (top-left), PointOdyssey (top-right, left), Dynamic Replica (top-right, right), and TAP-Vid sliding occluder (bottom) benchmarks**. Each model is trained in a shortened training setup. TAP-Vid results are measured in $\delta_{\text{avg}}^{\text{vis}}$ ($\uparrow$).

| Prior type | Kin. | RGB. | DAV. | Robo. | Avg. |
|---|---|---|---|---|---|
| Window-independent | 64.1 | **79.2** | 67.4 | 75.5 | 71.6 |
| Window-dependent | **66.5** | 77.9 | **74.2** | **78.0** | **74.1** |

| Prior type | PointOdy. | | Dyn. Rep. | |
|---|---|---|---|---|
| | $\delta_{\text{avg}}^{\text{vis}}$ | $\delta_{\text{avg}}^{\text{occ}}$ | $\delta_{\text{avg}}^{\text{vis}}$ | $\delta_{\text{avg}}^{\text{occ}}$ |
| Window-independent | 60.5 | 47.9 | 80.3 | 43.5 |
| Window-dependent | **64.5** | **51.6** | **81.9** | **48.1** |

| Prior type | Kinetics | | RGB-S | | DAVIS | | RoboTAP | | Avg. | |
|---|---|---|---|---|---|---|---|---|---|---|
| | $\delta_{\text{avg}}^{\text{vis}}$ | $\delta_{\text{avg}}^{\text{occ}}$ | $\delta_{\text{avg}}^{\text{vis}}$ | $\delta_{\text{avg}}^{\text{occ}}$ | $\delta_{\text{avg}}^{\text{vis}}$ | $\delta_{\text{avg}}^{\text{occ}}$ | $\delta_{\text{avg}}^{\text{vis}}$ | $\delta_{\text{avg}}^{\text{occ}}$ | $\delta_{\text{avg}}^{\text{vis}}$ | $\delta_{\text{avg}}^{\text{occ}}$ |
| Window-independent | 64.1 | 41.8 | **75.2** | 54.0 | 67.1 | 45.6 | 74.7 | 50.0 | 70.3 | 47.8 |
| Window-dependent | **65.9** | **50.9** | 73.7 | **59.2** | **73.6** | **59.0** | **76.4** | **61.9** | **72.4** | **57.8** |

## A.7 EFFECT OF VARIANCE SCHEDULE OF CONDITIONAL PROBABILITY PATH

In this experiment, we study the effect of the variance schedule used for the conditional probability path for point trajectories, visibilities, and confidences. We train three GenPT models in a shortened training setup on PointOdyssey. The first is our standard model, with the variance schedule: $l'^2\sigma_{\text{coord}}^2$, $l'^2\sigma_{\text{vis}}^2$, and $l'^2\sigma_{\text{conf}}^2$ for point trajectories, visibilities, and confidences, respectively. The second is a variant, with the variance schedule: $\sigma_{\text{coord}}^2$, $\sigma_{\text{vis}}^2$, and $\sigma_{\text{conf}}^2$ for point trajectories, visibilities, and confidences, respectively. The third is another variant, with the variance schedule set to zero for point trajectories, visibilities, and confidences.

Table 5 shows our quantitative results. Here we see that the model performs the worst when the variance is always set to zero, and performs the best when it's scaled according to the noise level $l$. Importantly, we found that there needs to be at least some noise in the sample during training, hence the large gap between the model with zero variance in its conditional probability path and the other two models, which have non-zero variance. However, we believe that a constant amount of variance in the conditional probability path can be detrimental as $l' \to 0$, since the noise in the conditional sample will increasingly compound with the noise in the prior sample, which may be too challenging for the model to estimate the ground truth from.

Table 5: **Effect of variance schedule of the conditional probability path on the TAP-Vid (top-left), PointOdyssey (top-right, left), Dynamic Replica (top-right, right), and TAP-Vid sliding occluder (bottom) benchmarks**. Each model is trained in a shortened training setup. TAP-Vid results are measured in $\delta_{\text{avg}}^{\text{vis}}$ ($\uparrow$).

| Variance sched. | Kin. | RGB. | DAV. | Robo. | Avg. |
|---|---|---|---|---|---|
| $0$ | 41.0 | 49.1 | 58.3 | 39.2 | 46.9 |
| $\sigma^2$ | 66.3 | 77.4 | 73.3 | 77.8 | 73.7 |
| $l'^2\sigma^2$ | **66.5** | **77.9** | **74.2** | **78.0** | **74.1** |

| Variance sched. | PointOdy. | | Dyn. Rep. | |
|---|---|---|---|---|
| | $\delta_{\text{avg}}^{\text{vis}}$ | $\delta_{\text{avg}}^{\text{occ}}$ | $\delta_{\text{avg}}^{\text{vis}}$ | $\delta_{\text{avg}}^{\text{occ}}$ |
| $0$ | 31.0 | 24.3 | 37.6 | 24.5 |
| $\sigma^2$ | **65.3** | **51.6** | **81.9** | 46.6 |
| $l'^2\sigma^2$ | 64.5 | **51.6** | **81.9** | **48.1** |

| Variance sched. | Kinetics | | RGB-S | | DAVIS | | RoboTAP | | Avg. | |
|---|---|---|---|---|---|---|---|---|---|---|
| | $\delta_{\text{avg}}^{\text{vis}}$ | $\delta_{\text{avg}}^{\text{occ}}$ | $\delta_{\text{avg}}^{\text{vis}}$ | $\delta_{\text{avg}}^{\text{occ}}$ | $\delta_{\text{avg}}^{\text{vis}}$ | $\delta_{\text{avg}}^{\text{occ}}$ | $\delta_{\text{avg}}^{\text{vis}}$ | $\delta_{\text{avg}}^{\text{occ}}$ | $\delta_{\text{avg}}^{\text{vis}}$ | $\delta_{\text{avg}}^{\text{occ}}$ |
| $0$ | 43.6 | 30.9 | 52.4 | 34.4 | 62.0 | 49.6 | 43.2 | 31.3 | 50.3 | 36.5 |
| $\sigma^2$ | **66.0** | 49.5 | 73.3 | **59.7** | 72.9 | 55.1 | **76.5** | 59.1 | 72.1 | 55.8 |
| $l'^2\sigma^2$ | 65.9 | **50.9** | **73.7** | 59.2 | **73.6** | **59.0** | 76.4 | **61.9** | **72.4** | **57.8** |

## A.8 SHOULD THERE BE ADDITIVE NOISE IN THE FIRST HALF OF THE SECOND PRIOR?

Previous work (Chen et al., 2024) has shown that it can be beneficial to add noise to previous samples when using them to initialize new samples. In our case, this would relate to the first half (along the temporal axis) of GenPT's second prior, where we have opted to use a Dirac delta function (*i.e.* no

noise) that is centred about the model's previous sample. Here we investigate the potential benefit of using a Gaussian instead of a Dirac delta function for the first half of the second prior. We train two GenPT models in a shortened training setup on PointOdyssey. The first is our standard model, which does not include noise in the first half of the second prior. The second is a variant, which includes noise in the first half of the second prior. Specifically, it includes the same additive noise used in the second half of the second prior.

Table 6 shows our quantitative results. Here we see that both models are almost equal when tracking visible points. When tracking occluded points, the model that does not include additional noise in the first half of its second prior is marginally better, which is our standard approach.

Table 6: **Effect of noise in first half of second prior on the TAP-Vid (top-left), PointOdyssey (top-right, left), Dynamic Replica (top-right, right), and TAP-Vid sliding occluder (bottom) benchmarks**. Each model is trained in a shortened training setup. TAP-Vid results are measured in $\delta_{avg}^{vis}$ ($\uparrow$).

| | Kin. | RGB. | DAV. | Robo. | Avg. | | PointOdy. $\delta_{avg}^{vis}$ | $\delta_{avg}^{occ}$ | Dyn. Rep. $\delta_{avg}^{vis}$ | $\delta_{avg}^{occ}$ |
|---|---|---|---|---|---|---|---|---|---|---|
| Includes noise | 66.1 | **79.4** | 72.9 | 77.9 | 74.1 | Includes noise | 64.3 | 50.2 | 81.5 | 47.0 |
| No noise | **66.5** | 77.9 | **74.2** | **78.0** | 74.1 | No noise | **64.5** | **51.6** | **81.9** | **48.1** |

| | Kinetics $\delta_{avg}^{vis}$ | $\delta_{avg}^{occ}$ | RGB-S $\delta_{avg}^{vis}$ | $\delta_{avg}^{occ}$ | DAVIS $\delta_{avg}^{vis}$ | $\delta_{avg}^{occ}$ | RoboTAP $\delta_{avg}^{vis}$ | $\delta_{avg}^{occ}$ | Avg. $\delta_{avg}^{vis}$ | $\delta_{avg}^{occ}$ |
|---|---|---|---|---|---|---|---|---|---|---|
| Includes noise | 65.7 | 49.4 | **75.1** | **59.7** | 72.2 | 55.7 | **76.7** | 59.8 | **72.5** | 56.1 |
| No noise | **65.9** | **50.9** | 73.7 | 59.2 | **73.6** | **59.0** | 76.4 | **61.9** | 72.4 | **57.8** |

### A.9    DISCRIMINATIVE *vs.* GENERATIVE TRAINING

In this experiment, we ablate the generative element of GenPT and observe the change to its tracking accuracy. Specifically, we remove the timestep conditioning; we remove the stochastic element from our two priors; and during training we always initialize trajectories, visibilities, and confidences from their associated (now deterministic) priors instead of sampling from a conditional distribution that's conditioned on a randomly sampled timestep. Integration is removed and the model simply takes $K = 6$ steps to arrive at an estimate of the ground truth. In other words, we have converted GenPT to follow the CoTracker3 training and testing setup as closely as possible. We train our models in a shortened training setup on PointOdyssey. The two types of GenPT models are labelled "Generative" (our main approach) and "Discriminative" (identical to CoTracker3's strategy and is deterministic).

Table 7 shows our quantitative results. Notably, we can see that the generative variant is consistently better than its discriminative counterpart by a wide margin.

Table 7: **Effect of discriminative *vs.* generative training on the TAP-Vid (top-left), PointOdyssey (top-right, left), Dynamic Replica (top-right, right), and TAP-Vid sliding occluder (bottom) benchmarks**. Each model is trained in a shortened training setup. TAP-Vid results are measured in $\delta_{avg}^{vis}$ ($\uparrow$).

| Model type | Kin. | RGB. | DAV. | Robo. | Avg. | Model type | PointOdy. $\delta_{avg}^{vis}$ | $\delta_{avg}^{occ}$ | Dyn. Rep. $\delta_{avg}^{vis}$ | $\delta_{avg}^{occ}$ |
|---|---|---|---|---|---|---|---|---|---|---|
| Discriminative | 66.0 | **79.0** | 72.6 | 76.7 | 73.6 | Discriminative | 63.4 | 50.2 | 80.5 | 45.5 |
| Generative | **66.5** | 77.9 | **74.2** | **78.0** | **74.1** | Generative | **64.5** | **51.6** | **81.9** | **48.1** |

| Model type | Kinetics $\delta_{avg}^{vis}$ | $\delta_{avg}^{occ}$ | RGB-S $\delta_{avg}^{vis}$ | $\delta_{avg}^{occ}$ | DAVIS $\delta_{avg}^{vis}$ | $\delta_{avg}^{occ}$ | RoboTAP $\delta_{avg}^{vis}$ | $\delta_{avg}^{occ}$ | Avg. $\delta_{avg}^{vis}$ | $\delta_{avg}^{occ}$ |
|---|---|---|---|---|---|---|---|---|---|---|
| Discriminative | 64.9 | 48.1 | **74.0** | 58.6 | 72.3 | 54.6 | 74.3 | 57.7 | 71.4 | 54.7 |
| Generative | **65.9** | **50.9** | 73.7 | **59.2** | **73.6** | **59.0** | **76.4** | **61.9** | **72.4** | **57.8** |

### A.10    EFFECT OF VARYING $\sigma_{coord}$

Here we study the effect on tracking accuracy when varying the standard deviation $\sigma_{coord}$ of the trajectory priors and conditional probability path. Table 8 shows our quantitative results. For each value of $\sigma_{coord}$ we train a separate model in a shortened training setup on PointOdyssey, where we

train for 100K steps and use a total batch size of two. In short summary, increasing $\sigma_{\text{coord}}$ results in lower tracking accuracy. For our main model, we chose $\sigma_{\text{coord}} = 0.25$ as we found it allows for a healthy amount of mode exploration during training while minimizing noise that may throw the tracker off track when the solution space is unimodal (*i.e.* when the point is easy to track).

Table 8: **Effect of varying $\sigma_{\text{coord}}$ on the TAP-Vid (top-left), PointOdyssey (top-right, left), Dynamic Replica (top-right, right), and TAP-Vid sliding occluder (bottom) benchmarks**. Each row contains results from a version of our model trained in a shortened training setup. TAP-Vid results are measured in $\delta_{\text{avg}}^{\text{vis}}$ ($\uparrow$).

| $\sigma_{\text{coord}}$ | Kinetics | RGB-S | DAVIS | RoboTAP | Avg. | $\sigma_{\text{coord}}$ | PointOdy. $\delta_{\text{avg}}^{\text{vis}}$ | $\delta_{\text{avg}}^{\text{occ}}$ | Dyn. Rep. $\delta_{\text{avg}}^{\text{vis}}$ | $\delta_{\text{avg}}^{\text{occ}}$ |
|---|---|---|---|---|---|---|---|---|---|---|
| 1.00 | 65.8 | **78.1** | 70.1 | 77.2 | 72.8 | 1.00 | 62.2 | 48.7 | 80.6 | 45.5 |
| 0.25 | **66.5** | 77.9 | **74.2** | **78.0** | **74.1** | 0.25 | **64.5** | **51.6** | **81.9** | **48.1** |

| $\sigma_{\text{coord}}$ | Kinetics $\delta_{\text{avg}}^{\text{vis}}$ | $\delta_{\text{avg}}^{\text{occ}}$ | RGB-S $\delta_{\text{avg}}^{\text{vis}}$ | $\delta_{\text{avg}}^{\text{occ}}$ | DAVIS $\delta_{\text{avg}}^{\text{vis}}$ | $\delta_{\text{avg}}^{\text{occ}}$ | RoboTAP $\delta_{\text{avg}}^{\text{vis}}$ | $\delta_{\text{avg}}^{\text{occ}}$ | Avg. $\delta_{\text{avg}}^{\text{vis}}$ | $\delta_{\text{avg}}^{\text{occ}}$ |
|---|---|---|---|---|---|---|---|---|---|---|
| 1.00 | 65.2 | 45.9 | **74.0** | 57.3 | 69.6 | 50.2 | 75.6 | 55.3 | 71.1 | 52.2 |
| 0.25 | **65.9** | **50.9** | 73.7 | **59.2** | **73.6** | **59.0** | **76.4** | **61.9** | **72.4** | **57.8** |

## A.11 REGRESSION *vs.* CLASSIFICATION-TYPE CONFIDENCE LOSS

Here we ablate the type of confidence loss used during training. Table 9 shows our quantitative results in a shortened training setup, as described previously. On average, the regression-based approach results in higher point tracking accuracy than the classification-based approach.

Table 9: **Effect of confidence loss type on the TAP-Vid (top-left), PointOdyssey (top-right, left), Dynamic Replica (top-right, right), and TAP-Vid sliding occluder (bottom) benchmarks**. Each model is trained in a shortened training setup. TAP-Vid results are measured in $\delta_{\text{avg}}^{\text{vis}}$ ($\uparrow$).

| Conf. loss type | Kin. | RGB. | DAV. | Robo. | Avg. | Conf. loss type | PointOdy. $\delta_{\text{avg}}^{\text{vis}}$ | $\delta_{\text{avg}}^{\text{occ}}$ | Dyn. Rep. $\delta_{\text{avg}}^{\text{vis}}$ | $\delta_{\text{avg}}^{\text{occ}}$ |
|---|---|---|---|---|---|---|---|---|---|---|
| Classification | 66.2 | 75.9 | **74.6** | 77.1 | 73.4 | Classification | 63.4 | 51.1 | 81.0 | **48.3** |
| Regression | **66.5** | **77.9** | 74.2 | **78.0** | **74.1** | Regression | **64.5** | **51.6** | **81.9** | 48.1 |

| Conf. loss type | Kinetics $\delta_{\text{avg}}^{\text{vis}}$ | $\delta_{\text{avg}}^{\text{occ}}$ | RGB-S $\delta_{\text{avg}}^{\text{vis}}$ | $\delta_{\text{avg}}^{\text{occ}}$ | DAVIS $\delta_{\text{avg}}^{\text{vis}}$ | $\delta_{\text{avg}}^{\text{occ}}$ | RoboTAP $\delta_{\text{avg}}^{\text{vis}}$ | $\delta_{\text{avg}}^{\text{occ}}$ | Avg. $\delta_{\text{avg}}^{\text{vis}}$ | $\delta_{\text{avg}}^{\text{occ}}$ |
|---|---|---|---|---|---|---|---|---|---|---|
| Classification | 65.7 | 50.9 | 73.4 | 58.7 | **73.9** | **59.5** | 75.6 | 60.8 | 72.1 | 57.4 |
| Regression | **65.9** | 50.9 | **73.7** | **59.2** | 73.6 | 59.0 | **76.4** | **61.9** | **72.4** | **57.8** |

## A.12 BEST OF N SAMPLES, GUIDED BY CONFIDENCE AND BEAM SEARCH

Here we continue our experiments from Sec. 4.4 and use beam search instead of greedy search (equivalent to beam search with a beam width of 1) when picking amongst the best of N sampled trajectories, guided by the model's confidence. Table 10 shows our quantitative results across all our benchmarks. In short, we find that beam search marginally improves upon greedy search. The marginal improvements at the cost of having to produce many times more samples leaves much to be desired. Based on the remaining performance gap when comparing to the oracle results (Table 1 and 2), we believe that improving the accuracy of the model's confidence predictions may widen the gap of the best of N results between using beam search and greedy search.

## A.13 COMPARING MODE EXPLORATION BETWEEN DISCRIMINATIVE AND GENERATIVE TRAINED MODELS

To further evaluate mode exploration, we compare GenPT with its discriminative counterpart, as introduced in Sec. A.9. Both models are trained in a shortened training setup, as described previously. The motivation of this experiment is to evaluate the impact of the generative training setup towards mode exploration capabilities during testing.

Table 10: **Best of N samples on the TAP-Vid (top-left), PointOdyssey (top-right, left), Dynamic Replica (top-right, right), and TAP-Vid sliding occluder (bottom) benchmarks, guided by confidence predictions and beam search**. For each sliding window, the best subset (with a size equal to the beam width) of N sampled trajectories is chosen, based on each trajectory's cumulative confidence score across all previous windows. At the final window, the trajectory with the highest cumulative confidence is chosen. TAP-Vid results are measured in $\delta_{avg}^{vis}$ ($\uparrow$).

| N | Kinetics | RGB-S | DAVIS | RoboTAP | Avg. |
|---|---|---|---|---|---|
| | | Beam width = 2 | | | |
| 5 | 68.7 | 82.2 | 78.4 | **80.6** | 77.5 |
| 10 | 68.9 | 82.2 | 78.5 | 80.5 | 77.5 |
| | | Beam width = 3 | | | |
| 5 | 68.9 | 82.2 | 78.5 | 80.5 | 77.5 |
| 10 | **69.0** | **82.4** | **78.6** | 80.5 | **77.6** |
| | | Beam width = 4 | | | |
| 5 | **69.0** | 82.2 | 78.2 | **80.6** | 77.5 |
| 10 | **69.0** | **82.4** | 78.4 | 80.5 | **77.6** |

| N | PointOdy. $\delta_{avg}^{vis}$ | $\delta_{avg}^{occ}$ | Dyn. Rep. $\delta_{avg}^{vis}$ | $\delta_{avg}^{occ}$ |
|---|---|---|---|---|
| | | Beam width = 2 | | |
| 5 | 71.9 | 58.4 | 85.7 | 54.1 |
| 10 | 71.8 | 58.5 | 85.7 | 54.2 |
| | | Beam width = 3 | | |
| 5 | 71.9 | 58.6 | 85.7 | 54.1 |
| 10 | **72.0** | 58.8 | 85.8 | 54.1 |
| | | Beam width = 4 | | |
| 5 | 71.8 | 58.5 | **85.8** | **54.3** |
| 10 | 71.9 | **58.8** | **85.8** | **54.3** |

| N | Kinetics $\delta_{avg}^{vis}$ | $\delta_{avg}^{occ}$ | RGB-S $\delta_{avg}^{vis}$ | $\delta_{avg}^{occ}$ | DAVIS $\delta_{avg}^{vis}$ | $\delta_{avg}^{occ}$ | RoboTAP $\delta_{avg}^{vis}$ | $\delta_{avg}^{occ}$ | Avg. $\delta_{avg}^{vis}$ | $\delta_{avg}^{occ}$ |
|---|---|---|---|---|---|---|---|---|---|---|
| | | | | | Beam width = 2 | | | | | |
| 5 | 68.4 | 56.0 | 78.9 | 69.6 | 78.0 | 67.1 | 79.2 | 69.1 | 76.1 | 65.5 |
| 10 | **68.5** | 56.1 | 79.2 | **70.1** | 78.0 | **67.2** | 79.2 | 69.0 | 76.2 | **65.6** |
| | | | | | Beam width = 3 | | | | | |
| 5 | **68.5** | **56.2** | 79.1 | 69.8 | **78.1** | 67.1 | 79.3 | **69.2** | 76.3 | **65.6** |
| 10 | **68.5** | 56.1 | **79.4** | **70.1** | **78.1** | 66.8 | 79.3 | 69.0 | 76.3 | 65.5 |
| | | | | | Beam width = 4 | | | | | |
| 5 | **68.5** | 56.1 | 79.0 | 69.9 | 77.8 | 67.0 | **79.4** | 69.1 | 76.2 | 65.5 |
| 10 | **68.5** | **56.2** | 79.3 | **70.1** | **78.1** | 66.9 | **79.4** | 69.0 | 76.3 | 65.5 |

Table 11: **Comparing tracking performance between discriminative and generative trained models on the TAP-Vid (left), PointOdyssey (middle, left), and Dynamic Replica (middle, right) benchmarks, as well as on a subset of the TAP-Vid sliding occluder (right) benchmark, when using the best of N samples, guided by an oracle**. For each sliding window, the best of N sampled trajectories is chosen, based on their distance to the ground truth. During test time, stochasticity is introduced into the discriminative model by sampling from the same priors used by the generative model. Each model is trained in a shortened training setup. TAP-Vid results are measured in $\delta_{avg}^{vis}$ ($\uparrow$).

| N | Kinetics | RGB-S | DAVIS | RoboTAP | Avg. |
|---|---|---|---|---|---|
| | | Generative model | | | |
| 1 | 66.5 | 77.9 | 74.2 | 78.0 | 74.1 |
| 5 | 69.3 | 81.2 | 76.7 | 80.7 | 77.0 |
| 10 | **70.1** | 82.3 | **77.3** | **81.6** | **77.8** |
| | Discriminative model w. test-time noise | | | | |
| 1 | 65.7 | 77.8 | 70.7 | 77.0 | 72.8 |
| 5 | 68.9 | 82.0 | 74.4 | 80.2 | 76.4 |
| 10 | 69.8 | **83.2** | 75.5 | 81.0 | 77.4 |

| N | PointOdy. $\delta_{avg}^{vis}$ | $\delta_{avg}^{occ}$ | Dyn. Rep. $\delta_{avg}^{vis}$ | $\delta_{avg}^{occ}$ |
|---|---|---|---|---|
| | | Generative model | | |
| 1 | 64.5 | 51.6 | 81.9 | 48.1 |
| 5 | 68.4 | 55.4 | 83.7 | 52.2 |
| 10 | **70.0** | **57.4** | **84.3** | **53.5** |
| | Discriminative model w. test-time noise | | | |
| 1 | 63.5 | 49.2 | 81.1 | 40.3 |
| 5 | 68.6 | 54.6 | 83.2 | 46.8 |
| 10 | **70.0** | 56.4 | 83.8 | 49.0 |

| N | RGB-S $\delta_{avg}^{vis}$ | $\delta_{avg}^{occ}$ | DAVIS $\delta_{avg}^{vis}$ | $\delta_{avg}^{occ}$ |
|---|---|---|---|---|
| | | Generative model | | |
| 1 | 73.7 | 59.2 | 73.6 | 59.2 |
| 5 | 78.1 | 66.5 | 76.1 | 66.5 |
| 10 | 79.6 | **68.9** | **77.0** | **68.9** |
| | Discriminative model w. test-time noise | | | |
| 1 | 73.6 | 48.4 | 70.3 | 44.9 |
| 5 | 78.4 | 58.1 | 74.1 | 54.9 |
| 10 | **79.6** | 60.6 | 75.0 | 57.8 |

Tables 11 and 12 show quantitative results in the best of N setup. To allow the discriminative model to have a varying set of trajectories to choose from, we introduce stochasticity into its trajectory, visibility, and confidence initialization stage by sampling from the same priors used by the generative model. Although the discriminative model can leverage its confidence estimates to choose the best of N stochastically-initialized samples to improve its tracking accuracy, we can see that overall, its tracking accuracy is lower than the generative model's tracking accuracy, especially for occluded points. This may indicate that the generative model—trained to map random samples from a conditional prior to plausible point trajectories—is more capable at capturing meaningful modes in areas of uncertainty.

To further validate this statement, we experiment with a worst of N setup (oracle only), where both the minimally-trained generative model and its discriminative variant (using the generative priors at

Table 12: **Comparing tracking performance between discriminative and generative trained models on the TAP-Vid (left), PointOdyssey (middle, left), and Dynamic Replica (middle, right) benchmarks, as well as on a subset of the TAP-Vid sliding occluder (right) benchmark, when using the best of N samples, guided by confidence predictions and greedy search**. For each sliding window, the best of N sampled trajectories is chosen, based on the model's confidence prediction for each trajectory. During test time, stochasticity is introduced into the discriminative model by sampling from the same priors used by the generative model. Each model is trained in a shortened training setup. TAP-Vid results are measured in $\delta_{avg}^{vis}$ (↑).

| N | Kinetics | RGB-S | DAVIS | RoboTAP | Avg. |
|---|---|---|---|---|---|
| | | Generative model | | | |
| 1 | 66.5 | 77.9 | 74.2 | 78.0 | 74.1 |
| 5 | 67.3 | 79.2 | **74.6** | 78.8 | 75.0 |
| 10 | **67.4** | **79.4** | 74.5 | **78.9** | **75.1** |
| | Discriminative model w. test-time noise | | | | |
| 1 | 65.7 | 77.8 | 70.7 | 77.0 | 72.8 |
| 5 | 66.3 | 78.7 | 71.8 | 77.6 | 73.6 |
| 10 | 66.5 | 78.8 | 72.3 | 77.7 | 73.8 |

| | PointOdy. | | Dyn. Rep. | | | RGB-S | | DAVIS | |
|---|---|---|---|---|---|---|---|---|---|
| N | $\delta_{avg}^{vis}$ | $\delta_{avg}^{occ}$ | $\delta_{avg}^{vis}$ | $\delta_{avg}^{occ}$ | N | $\delta_{avg}^{vis}$ | $\delta_{avg}^{occ}$ | $\delta_{avg}^{vis}$ | $\delta_{avg}^{occ}$ |
| | Generative model | | | | | Generative model | | | |
| 1 | 64.5 | 51.6 | 81.9 | 48.1 | 1 | 73.7 | 59.2 | 73.6 | 59.2 |
| 5 | 65.8 | 52.4 | 82.2 | 48.5 | 5 | 75.2 | 61.7 | 74.0 | 60.6 |
| 10 | **65.9** | **52.6** | **82.3** | **48.7** | 10 | **75.5** | **62.1** | **73.9** | **62.1** |
| | Discriminative model w. test-time noise | | | | | Discriminative model w. test-time noise | | | |
| 1 | 63.5 | 49.2 | 81.1 | 40.3 | 1 | 73.6 | 48.4 | 70.3 | 44.9 |
| 5 | 64.2 | 50.0 | 81.2 | 41.4 | 5 | 74.3 | 50.4 | 71.1 | 47.1 |
| 10 | 64.0 | 49.6 | 81.3 | 41.2 | 10 | 74.4 | 50.8 | 71.4 | 48.0 |

Table 13: **Comparing tracking performance between discriminative and generative trained models on the TAP-Vid (left), PointOdyssey (middle, left), and Dynamic Replica (middle, right) benchmarks, as well as on a subset of the TAP-Vid sliding occluder (right) benchmark, when using the worst of N samples, guided by an oracle**. For each sliding window, the worst of N sampled trajectories is chosen, based on their distance to the ground truth. During test time, stochasticity is introduced into the discriminative model by sampling from the same priors used by the generative model. Each model is trained in a shortened training setup. TAP-Vid results are measured in $\delta_{avg}^{vis}$ (↑).

| N | Kinetics | RGB-S | DAVIS | RoboTAP | Avg. |
|---|---|---|---|---|---|
| | | Generative model | | | |
| 1 | **66.5** | **77.9** | **74.2** | **78.0** | **74.1** |
| 5 | 63.6 | 73.6 | 71.6 | 74.1 | 70.7 |
| 10 | 62.6 | 72.1 | 70.5 | 72.4 | 69.4 |
| | Discriminative model w. test-time noise | | | | |
| 1 | 65.7 | 77.8 | 70.7 | 77.0 | 72.8 |
| 5 | 61.5 | 71.9 | 66.6 | 70.8 | 67.7 |
| 10 | 59.5 | 69.1 | 65.1 | 67.6 | 65.3 |

| | PointOdy. | | Dyn. Rep. | | | RGB-S | | DAVIS | |
|---|---|---|---|---|---|---|---|---|---|
| N | $\delta_{avg}^{vis}$ | $\delta_{avg}^{occ}$ | $\delta_{avg}^{vis}$ | $\delta_{avg}^{occ}$ | N | $\delta_{avg}^{vis}$ | $\delta_{avg}^{occ}$ | $\delta_{avg}^{vis}$ | $\delta_{avg}^{occ}$ |
| | Generative model | | | | | Generative model | | | |
| 1 | **64.5** | **51.6** | **81.9** | **48.1** | 1 | **73.7** | **59.2** | **73.6** | **59.2** |
| 5 | 60.8 | 48.0 | 80.1 | 44.7 | 5 | 68.6 | 52.1 | 69.9 | 53.8 |
| 10 | 59.5 | 46.9 | 79.6 | 43.7 | 10 | 66.6 | 49.4 | 68.8 | 51.9 |
| | Discriminative model w. test-time noise | | | | | Discriminative model w. test-time noise | | | |
| 1 | 63.5 | 49.2 | 81.1 | 40.3 | 1 | 73.6 | 48.4 | 70.3 | 44.9 |
| 5 | 57.4 | 42.8 | 78.0 | 33.4 | 5 | 66.9 | 38.1 | 65.1 | 35.1 |
| 10 | 54.5 | 39.8 | 76.7 | 31.2 | 10 | 63.7 | 34.3 | 63.1 | 31.3 |

test-time) pick the worst of N trajectories during the sliding window process based on an oracle, in test time. The intent of the experiment is to highlight the difference in variance in point trajectories between the generative model and its discriminative variant, specifically on occluded points. The quantitative results are shown in Table 13. As N increases, the discriminative variant experiences a much larger drop in point tracking performance compared to the generative model, which is even more pronounced on occluded points. In other words, the generative model is generating trajectories that lie closer to the true data distribution than the discriminative variant, especially when the points are occluded.

We provide qualitative evidence in Fig. 6 and 7, where we show a visual comparison of 100 sampled point trajectories (tracking the same point) generated by each model. Without generative training, the discriminative model often fails to capture meaningful modes in areas of uncertainty when the generative prior is used during test time. In contrast, the generative model is more capable at capturing meaningful modes in areas of uncertainty.

### A.14 TEST-TIME APPLICATION OF THE GENERATIVE PRIOR IN COTRACKER3

Here we extend upon the experiments in Sec. A.13 by providing a comparison of best of N and worst of N results between our best GenPT model (trained on PointOdyssey) and the strongest competing pre-trained CoTracker3 model (pre-trained on Kub+15k). Since CoTracker3 is a discriminative model, during test time we introduce stochasticity into its trajectory, visibility, and confidence initialization by sampling from the same generative priors used by GenPT. The intent of this experiment is to show that GenPT's generative training—trained to map random samples from a conditional prior to plausible point trajectories—results in a model that captures meaningful modes in areas of uncertainty, while adding random perturbations to a discriminative model in test-time does not.

Tables 14 and 15 show quantitative results in the best of N setup, while Table 16 shows quantitative results in the worst of N setup (oracle only). Similar to what was shown in Sec. A.13, in general there is a substantial gap in tracking accuracy between the generative model (GenPT) and its discriminative competition (in this case, CoTracker3), which is more pronounced on occluded points. As N increases in the worst of N setup, CoTracker3 experiences a much larger drop in point tracking performance compared to GenPT, especially on occluded points. This experiment provides further evidence that GenPT's generative training results in a generative model that captures meaningful modes in areas of uncertainty, while adding random perturbations to a discriminative model in test-time does not.

Table 14: **Comparing tracking performance between the best CoTracker3 model (pre-trained on Kub+15k) and our best GenPT model (trained on PointOdyssey) on a subset of the TAP-Vid benchmark (left), PointOdyssey (middle, left), and Dynamic Replica (middle, right) benchmarks, as well as on a subset of the TAP-Vid sliding occluder (right) benchmark, when using the best of N samples, guided by an oracle**. For each sliding window, the best of N sampled trajectories is chosen, based on their distance to the ground truth. During test time, stochasticity is introduced into CoTracker3 by sampling from the same priors used by GenPT. TAP-Vid results are measured in $\delta^{\text{vis}}_{\text{avg}}$ ($\uparrow$).

| N | RGB-S | DAVIS | N | PointOdy. $\delta^{\text{vis}}_{\text{avg}}$ | $\delta^{\text{occ}}_{\text{avg}}$ | Dyn. Rep. $\delta^{\text{vis}}_{\text{avg}}$ | $\delta^{\text{occ}}_{\text{avg}}$ | N | RGB-S $\delta^{\text{vis}}_{\text{avg}}$ | $\delta^{\text{occ}}_{\text{avg}}$ | DAVIS $\delta^{\text{vis}}_{\text{avg}}$ | $\delta^{\text{occ}}_{\text{avg}}$ |
|---|---|---|---|---|---|---|---|---|---|---|---|---|
| | GenPT (PO) | | | GenPT (PO) | | | | | GenPT (PO) | | | |
| 1 | 79.4 | 77.7 | 1 | 70.2 | 57.9 | 85.4 | 53.8 | 1 | 76.1 | 66.4 | 77.6 | 66.5 |
| 5 | 84.3 | 80.1 | 5 | 74.0 | 62.7 | 87.3 | 57.7 | 5 | 82.4 | 74.4 | 80.0 | 70.3 |
| 10 | **85.8** | **80.5** | 10 | **75.1** | **64.2** | **88.0** | **59.2** | 10 | **84.3** | **76.9** | **80.5** | **71.1** |
| | CoTracker3 (Kub+15k) w. test-time noise | | | CoTracker3 (Kub+15k) w. test-time noise | | | | | CoTracker3 (Kub+15k) w. test-time noise | | | |
| 1 | 73.4 | 71.7 | 1 | 56.2 | 42.1 | 80.5 | 31.8 | 1 | 67.8 | 17.8 | 69.6 | 31.1 |
| 5 | 80.4 | 76.2 | 5 | 65.9 | 52.1 | 83.2 | 37.0 | 5 | 74.5 | 24.5 | 75.3 | 44.1 |
| 10 | 82.2 | 77.3 | 10 | 68.7 | 54.9 | 83.9 | 38.7 | 10 | 76.1 | 26.8 | 76.3 | 46.7 |

Table 15: **Comparing tracking performance between the best CoTracker3 model (pre-trained on Kub+15k) and our best GenPT model (trained on PointOdyssey) on a subset of the TAP-Vid benchmark (left), PointOdyssey (middle, left), and Dynamic Replica (middle, right) benchmarks, as well as on a subset of the TAP-Vid sliding occluder (right) benchmark, when using the best of N samples, guided by confidence predictions and greedy search**. For each sliding window, the best of N sampled trajectories is chosen, based on the model's confidence prediction for each trajectory. During test time, stochasticity is introduced into CoTracker3 by sampling from the same priors used by GenPT. TAP-Vid results are measured in $\delta^{\text{vis}}_{\text{avg}}$ ($\uparrow$).

| N | RGB-S | DAVIS | N | PointOdy. $\delta^{\text{vis}}_{\text{avg}}$ | $\delta^{\text{occ}}_{\text{avg}}$ | Dyn. Rep. $\delta^{\text{vis}}_{\text{avg}}$ | $\delta^{\text{occ}}_{\text{avg}}$ | N | RGB-S $\delta^{\text{vis}}_{\text{avg}}$ | $\delta^{\text{occ}}_{\text{avg}}$ | DAVIS $\delta^{\text{vis}}_{\text{avg}}$ | $\delta^{\text{occ}}_{\text{avg}}$ |
|---|---|---|---|---|---|---|---|---|---|---|---|---|
| | GenPT (PO) | | | GenPT (PO) | | | | | GenPT (PO) | | | |
| 1 | 79.4 | 77.7 | 1 | 70.2 | 57.9 | 85.4 | 53.8 | 1 | 76.1 | 66.4 | 77.6 | 66.5 |
| 5 | 81.7 | **78.3** | 5 | 71.4 | **58.9** | **85.7** | **54.2** | 5 | 78.7 | 69.4 | 77.7 | 66.9 |
| 10 | **82.2** | **78.3** | 10 | **71.6** | 58.8 | 85.6 | **54.2** | 10 | **79.1** | **69.8** | **77.9** | **67.6** |
| | CoTracker3 (Kub+15k) w. test-time noise | | | CoTracker3 (Kub+15k) w. test-time noise | | | | | CoTracker3 (Kub+15k) w. test-time noise | | | |
| 1 | 73.4 | 71.7 | 1 | 56.2 | 42.1 | 80.5 | 31.8 | 1 | 67.8 | 17.8 | 69.6 | 31.1 |
| 5 | 77.5 | 74.8 | 5 | 60.9 | 46.6 | 81.8 | 32.5 | 5 | 71.0 | 19.2 | 72.9 | 35.9 |
| 10 | 78.4 | 75.1 | 10 | 62.0 | 47.4 | 81.9 | 32.7 | 10 | 71.9 | 19.5 | 73.3 | 36.3 |

Table 16: **Comparing tracking performance between the best CoTracker3 model (pre-trained on Kub+15k) and our best GenPT model (trained on PointOdyssey) on a subset of the TAP-Vid benchmark (left), PointOdyssey (middle, left), and Dynamic Replica (middle, right) benchmarks, as well as on a subset of the TAP-Vid sliding occluder (right) benchmark, when using the worst of N samples, guided by an oracle**. For each sliding window, the worst of N sampled trajectories is chosen, based on their distance to the ground truth. During test time, stochasticity is introduced into CoTracker3 by sampling from the same priors used by GenPT. TAP-Vid results are measured in $\delta^{vis}_{avg}$ ($\uparrow$).

| N | RGB-S | DAVIS |
|---|---|---|
| | GenPT (PO) | |
| 1 | **79.4** | **77.7** |
| 5 | 73.5 | 75.3 |
| 10 | 71.9 | 74.3 |
| | CoTracker3 (Kub+15k) w. test-time noise | |
| 1 | 73.4 | 71.7 |
| 5 | 65.0 | 64.1 |
| 10 | 61.1 | 60.4 |

| | PointOdy. | | Dyn. Rep. | |
| N | $\delta^{vis}_{avg}$ | $\delta^{occ}_{avg}$ | $\delta^{vis}_{avg}$ | $\delta^{occ}_{avg}$ |
|---|---|---|---|---|
| | GenPT (PO) | | | |
| 1 | **70.2** | **57.9** | **85.4** | **53.8** |
| 5 | 66.6 | 53.4 | 83.1 | 49.7 |
| 10 | 65.6 | 52.4 | 82.2 | 48.5 |
| | CoTracker3 (Kub+15k) w. test-time noise | | | |
| 1 | 56.2 | 42.1 | 80.5 | 31.8 |
| 5 | 48.1 | 33.8 | 76.5 | 27.1 |
| 10 | 45.4 | 31.8 | 74.4 | 25.6 |

| | RGB-S | | DAVIS | |
| N | $\delta^{vis}_{avg}$ | $\delta^{occ}_{avg}$ | $\delta^{vis}_{avg}$ | $\delta^{occ}_{avg}$ |
|---|---|---|---|---|
| | GenPT (PO) | | | |
| 1 | **76.1** | **66.4** | **77.6** | **66.5** |
| 5 | 69.2 | 57.2 | 74.2 | 62.4 |
| 10 | 66.9 | 53.9 | 73.1 | 60.4 |
| | CoTracker3 (Kub+15k) w. test-time noise | | | |
| 1 | 67.8 | 17.8 | 69.6 | 31.1 |
| 5 | 59.2 | 12.2 | 61.4 | 18.2 |
| 10 | 55.7 | 10.5 | 58.1 | 14.5 |

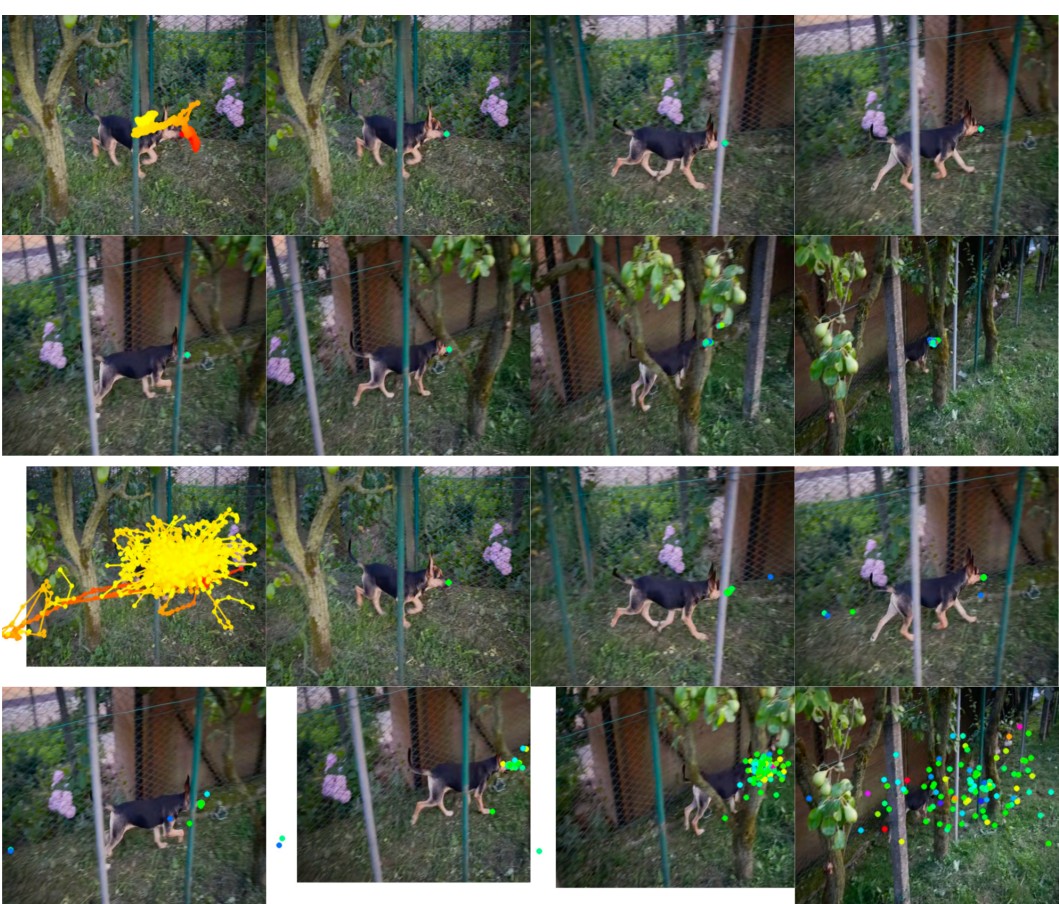

Figure 6: **Comparing multi-modality between discriminative and generative trained models (1/2)**. 100 randomly sampled trajectories of a single tracked point. The first two rows corresponds to GenPT trained and tested with its typical generative setup. The second two rows corresponds to a discriminative variant of GenPT trained with CoTracker3's initialization strategy (*i.e.* deterministic) and tested using GenPT's generative prior. The first image of each two-row block shows the full path of each sampled trajectory. This experiment shows how introducing stochasticity to a discriminative trained model during testing is insufficient in capturing meaningful variance in areas of uncertainty— generative training is required. Video is from the TAP-Vid DAVIS dataset (Doersch et al., 2022).

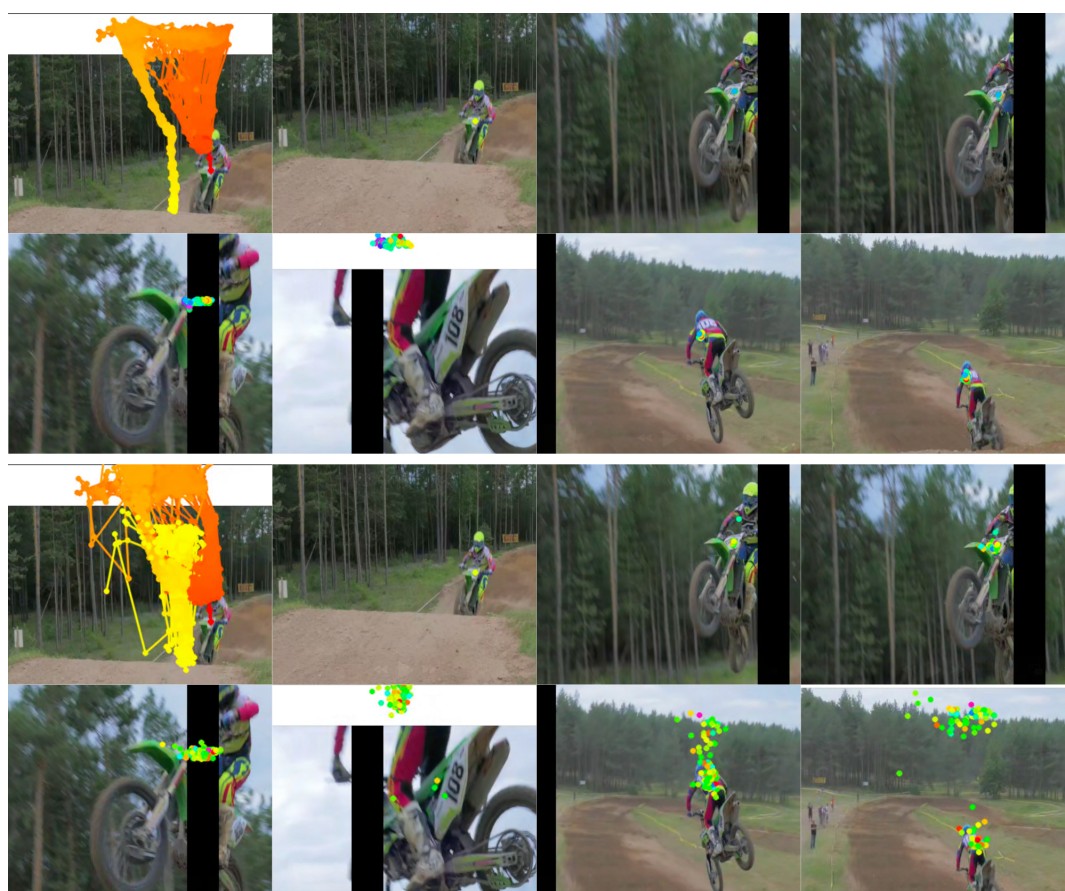

Figure 7: **Comparing multi-modality between discriminative and generative trained models (2/2)**. 100 randomly sampled trajectories of a single tracked point. The first two rows corresponds to GenPT trained and tested with its typical generative setup. The second two rows corresponds to a discriminative variant of GenPT trained with CoTracker3's initialization strategy (*i.e.* deterministic) and tested using GenPT's generative prior. The first image of each two-row block shows the full path of each sampled trajectory. This experiment shows how introducing stochasticity to a discriminative trained model during testing is insufficient in capturing meaningful variance in areas of uncertainty—generative training is required. Video is from the TAP-Vid DAVIS dataset (Doersch et al., 2022).

## A.15 CONVERGENCE STABILITY

Fig. 8 compares the stability in point tracking accuracy over the number of iterative updates (during inference) between GenPT and the CoTracker3 model we trained. The results are averaged over the PointOdyssey and Dynamic Replica benchmarks. Our model maintains a higher tracking accuracy over the number of steps, degrading at a slower rate than CoTracker3.

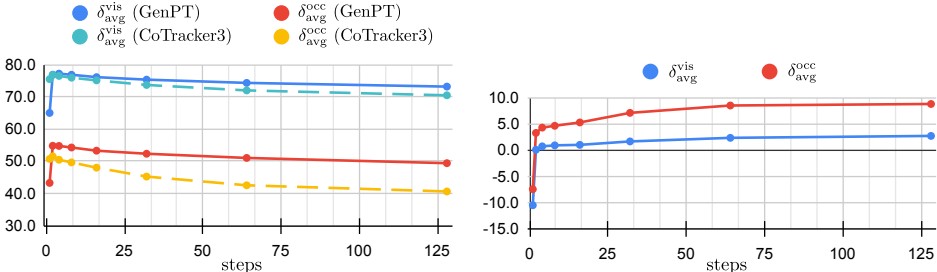

Figure 8: **Convergence stability**. (Left) Tracking accuracy on PointOdyssey and Dynamic Replica benchmarks (averaged over both test sets) after taking refinement steps in powers of two (up to 128 steps). Here we use one integration step for our model, instead of the usual two integration steps. (Right) Difference in tracking accuracy between our model and CoTracker3. The increase in difference over steps shows that our model's tracking accuracy over the number of refinement steps degrades at a slower rate than CoTracker3's.

## A.16 RUNTIME ANALYSIS

Here we compare GenPT's runtime performance with CoTracker3. We follow the same protocol used by the authors of CoTracker3, where we measure the average time it takes for the method to process one frame (in seconds), with the number of tracked points varying between 1 and 10,000. We average this across all videos in the DAVIS test set. We include an offline variant of CoTracker3, where the entire video is processed at once instead of in a sliding window fashion (online), effectively reducing the total number of function evaluations by half. We also report runtime performance when using the best-of-5 and best-of-10 samples, guided by GenPT's confidence. Some experiments max out at a lower number of tracked points due to running out of memory.

Fig. 9 shows our results. GenPT is around twice as fast as CoTracker3. Unfortunately, using the best-of-N samples can dramatically increase GenPT's runtime, since each window contains at least N times the number of function evaluations, which includes the additional overhead in picking the best of those samples through confidence comparisons and gather operations. When cross-referencing with Table 1 and 2, one can see that there is a substantial cost/benefit trade-off of best-of-N in terms of runtime *vs*. accuracy. When considering the gap between oracle and greedy performance and the leanness of GenPT's architecture, we believe the cost/benefit trade-off of best-of-N can be best addressed by improving GenPT's confidence accuracy.

## A.17 VARYING $K$ AND $L$ DURING INFERENCE

We experiment with varying the number of ground truth estimates and integration steps during inference with our PointOdyssey-trained GenPT model. Note that the model is still trained with $K = 4$. Our quantitative results on the PointOdyssey and Dynamic Replica benchmarks are shown in Table 17. Fig. 10 shows an example of the sampling process alongside ground truth estimates for 100 randomly sampled trajectories of a single tracked point.

## A.18 LIMITATIONS.

A primary limitation of GenPT, shared with other point trackers, is the high memory and compute cost of training due to its reliance on iterative refinements and dense correlation features. This poses a challenge for scaling the model to fully explore its generative potential. While flow matching is typically "simulation-free," our use of iterative refinements during training reintroduces some of the computational complexity that flow matching aims to avoid. This presents a clear direction for future

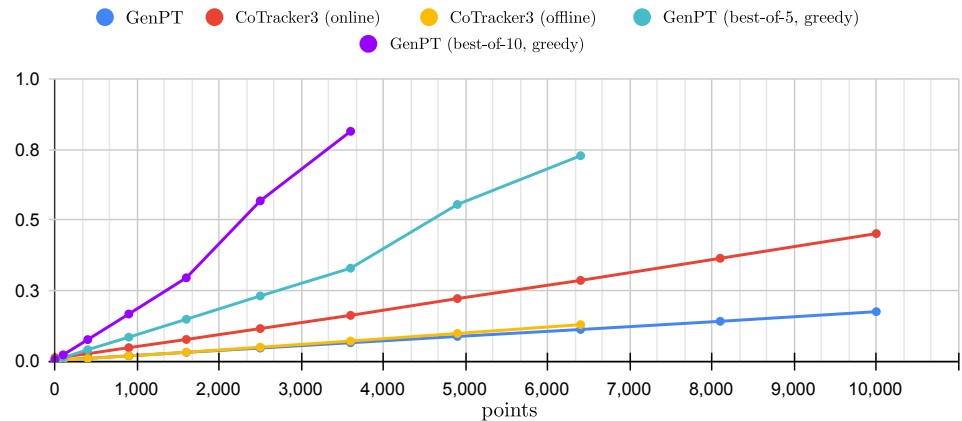

Figure 9: **Runtime analysis**. We evaluate the speed of GenPT and CoTracker3 on the TAP-Vid DAVIS test set (Doersch et al., 2022) and report the average time (in seconds) it takes to process a frame when tracking 1 to 10,000 points. We include an offline variant of CoTracker3, where the entire video is processed at once instead of in a sliding window fashion (online), effectively reducing the total number of function evaluations by half. We also report runtime performance when using the best-of-5 and best-of-10 samples, guided by GenPT's confidence. Some experiments max out at a lower number of tracked points due to running out of memory. GenPT is around twice as fast as CoTracker3. Using the best-of-N samples can dramatically increase the runtime, since each window contains N times the number of function evaluations.

Table 17: **Effect of varying $K$ and $L$ during inference on the PointOdyssey and Dynamic Replica benchmarks**.

| | PointOdy. | | Dyn. Rep. | | Avg. | |
|---|---|---|---|---|---|---|
| | $\delta^{vis}_{avg}$ | $\delta^{occ}_{avg}$ | $\delta^{vis}_{avg}$ | $\delta^{occ}_{avg}$ | $\delta^{vis}_{avg}$ | $\delta^{occ}_{avg}$ |
| $K=3, L=2$ | 69.4 | 56.7 | 84.9 | 52.8 | 77.1 | 54.8 |
| $K=3, L=3$ | **70.2** | **57.9** | **85.4** | **53.8** | **77.8** | **55.9** |
| $K=3, L=4$ | 69.9 | 57.5 | 85.2 | 53.3 | 77.5 | 55.4 |
| $K=4, L=2$ | 69.6 | 56.8 | 84.8 | 52.7 | 77.2 | 54.7 |
| $K=4, L=3$ | **70.2** | 57.8 | 85.3 | 53.7 | **77.8** | 55.8 |
| $K=4, L=4$ | 69.8 | 57.5 | 85.1 | 53.4 | 77.4 | 55.5 |
| $K=5, L=2$ | 69.4 | 56.6 | 84.7 | 52.6 | 77.1 | 54.6 |
| $K=5, L=3$ | 70.1 | 57.5 | 85.3 | 53.5 | 77.7 | 55.5 |
| $K=5, L=4$ | 69.8 | 57.1 | 85.1 | 53.4 | 77.4 | 55.3 |

work: developing a more efficient training strategy. Another avenue for improvement is to extend GenPT's capabilities to forecast and backcast trajectories beyond its current window. This could be achieved by training the model in a masked manner or by adapting it from large video diffusion models that are already capable of generating future frames. These enhancements could make GenPT valuable for applications requiring predictive capabilities, such as collision detection.

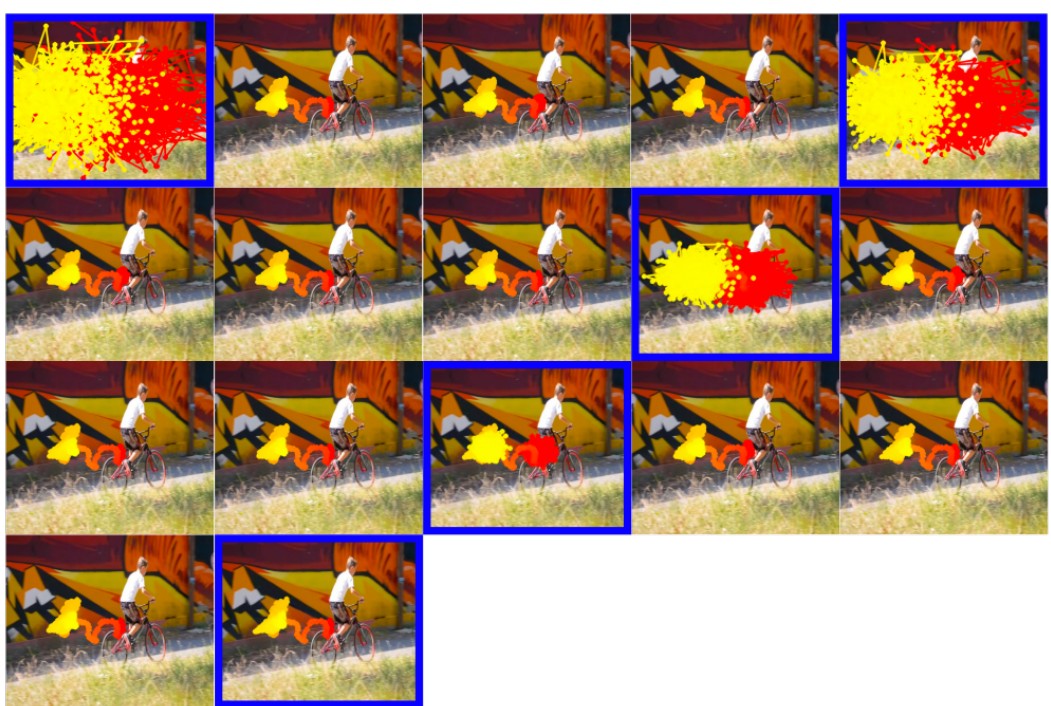

Figure 10: **Visualizing sample integration alongside ground truth estimates**. Shown above is the iterative refinement of ground truth estimates alongside samples generated through ODE integration with GenPT for 100 randomly sampled trajectories of a single tracked point. Images outlined in blue show a sample, while other images show a ground truth estimate. Starting from the prior (top-left), GenPT takes $K$ refinement steps towards an estimate of the ground truth ($K = 3$ for this figure). With the $K^{\text{th}}$ ground truth estimate, GenPT produces the velocity field needed to integrate to the next sample through Euler integration. The final output is the sample generated after repeating the process $L - 1$ times (bottom-right), where $L = 5$ for this figure. Video is from the TAP-Vid DAVIS dataset (Doersch et al., 2022).

---

**Algorithm 1:** One training step of GenPT

**Input** : Transformer $\mathcal{T}_\theta$, encoder $\mathcal{F}_\phi$, MLP $\mathcal{M}_\rho$, tuple $\{I, P, V\}$.

**Output** : Training loss $\mathcal{L}$.

**Note** : Assume tensor broadcasting when no explicit repeats are performed. $\mathtt{cat}(\{\cdot, \ldots\}, n)$ concatenates a tuple of tensors along the $n^{\text{th}}$ axis. $\mathtt{slice(I, T, n)}$ slices tensor $I$ into $N_T$ overlapping windows of size $T$ along its $n^{\text{th}}$ axis.

1   $P^q \in \mathbb{R}^{N_{Pq} \times 3} \leftarrow$ first visible points from each trajectory in $P$

2   $P^q \leftarrow \mathtt{repeat}(P^q, \text{``}N_{Pq}\ 3 \rightarrow T\ N_{Pq}\ 3\text{''}) \in \mathbb{R}^{T \times N_{Pq} \times 3}$

3   $\{F^s \in \mathbb{R}^{T' \times D_\mathcal{F} \times H/2^{s+1} \times W/2^{s+1}}\}_{s=1}^S \leftarrow \mathtt{get\_fmaps\_pyramid}(\mathcal{F}_\phi, I, S)$

    `// Sample multiscale feature crops centred at query points.`

4   $f^{\Omega(P^q)} \leftarrow \mathtt{sample\_feats}(\{F^s\}_{s=1}^S, \Omega(P^q)) \in \mathbb{R}^{S \times T \times N_{Pq} \times D_\mathcal{F} \times 49}$

    `// Slice into` $N_T$ `overlapping windows of size` $T$`.`

5   $\{P, V\} \leftarrow \{\mathtt{slice}(P, T, 1) \in \mathbb{R}^{N_T \times T \times N_{Pq} \times 2}, \mathtt{slice}(V, T, 1) \in \mathbb{R}^{N_T \times T \times N_{Pq} \times 1}\}$

6   $\{F^s\}_{s=1}^S \leftarrow \{\mathtt{slice}(F^s, T, 2) \in \mathbb{R}^{N_T \times T \times D_\mathcal{F} \times H/2^{s+1} \times W/2^{s+1}}\}_{s=1}^S$

7   $\mathcal{L} \leftarrow 0$

    `// Initialize ground truth estimates.`

8   $\{\hat{P}, \hat{V}, \hat{C}\} \leftarrow \{\mathbf{0} \in \mathbb{R}^{N_T \times K \times L \times T \times N_{Pq} \times 2}, \mathbf{0} \in \mathbb{R}^{N_T \times K \times L \times T \times N_{Pq} \times 1}, \mathbf{0} \in \mathbb{R}^{N_T \times K \times L \times T \times N_{Pq} \times 1}\}$

    `// Initialize samples.`

9   $\{\tilde{P}, \tilde{V}, \tilde{C}\} \leftarrow \{\mathbf{0} \in \mathbb{R}^{N_T \times L \times T \times N_{Pq} \times 2}, \mathbf{0} \in \mathbb{R}^{N_T \times L \times T \times N_{Pq} \times 1}, \mathbf{0} \in \mathbb{R}^{N_T \times L \times T \times N_{Pq} \times 1}\}$

    `// Initialize ground truth confidences.`

10   $C \leftarrow \mathbf{0} \in \mathbb{R}^{N_T \times K \times L \times T \times N_{Pq} \times 1}$

11   **for** $n_T \leftarrow 1$ **to** $N_T$ **do**

12     $\mathtt{frame\_inds} \leftarrow \mathtt{repeat}(\mathtt{cat}(\{t\}_{t=0}^{T-1}, 1), \text{``}T \rightarrow T\ N_{Pq}\ 1\text{''}) \in \mathbb{R}^{T \times N_{Pq} \times 1}$

13     $\{\epsilon_{\text{coord}}, \epsilon_{\text{vis}}, \epsilon_{\text{conf}}\} \sim \{\mathcal{N}(0, I), \mathcal{N}(0, I), \mathcal{N}(0, I)\}$

14     **if** $n_T = 1$ **then**

        `// In first window. Sample from first prior.`

15       $\{\tilde{P}_{n_T, 1}, \tilde{V}_{n_T, 1}, \tilde{C}_{n_T, 1}\} \leftarrow \{P^q + \sigma_{\text{coord}}\epsilon_{\text{coord}}, \sigma_{\text{vis}}\epsilon_{\text{vis}}, \sigma_{\text{conf}}\epsilon_{\text{conf}}\}$

16       $\mathtt{prior\_flag} \leftarrow \mathbf{1} \in \mathbb{R}^{T \times N_{Pq} \times 1}$

17     **else**

        `// Not in first window. Sample from second prior.`

18       $\{\tilde{P}_{n_T, 1, 1:(T/2)}, \tilde{P}_{n_T, 1, (T/2+1):T}\} \leftarrow \{\hat{P}_{(n_T-1), K, l, (T/2+1):T}, \hat{P}_{(n_T-1), K, l, T} + \sigma_{\text{coord}}\epsilon_{\text{coord}}\}$

19       $\{\tilde{V}_{n_T, 1, 1:(T/2)}, \tilde{V}_{n_T, 1, (T/2+1):T}\} \leftarrow \{\hat{V}_{(n_T-1), K, l, (T/2+1):T}, \hat{V}_{(n_T-1), K, l, T} + \sigma_{\text{vis}}\epsilon_{\text{vis}}\}$

20       $\{\tilde{C}_{n_T, 1, 1:(T/2)}, \tilde{C}_{n_T, 1, (T/2+1):T}\} \leftarrow \{\hat{C}_{(n_T-1), K, l, (T/2+1):T}, \hat{C}_{(n_T-1), K, l, T} + \sigma_{\text{conf}}\epsilon_{\text{conf}}\}$

21       $\mathtt{prior\_flag} \leftarrow \mathbf{0} \in \mathbb{R}^{T \times N_{Pq} \times 1}$

22     **end**

23     $l \sim \mathcal{U}\{1, L\}$                                                 `// Sample an ODE timestep.`

24     $l' \leftarrow \mathtt{repeat}(l, \text{``}1 \rightarrow T\ N_{Pq}\ 1\text{''}) \in \mathbb{R}^{T \times N_{Pq} \times 1}$

25     $l' \leftarrow (l' - 1)/(L - 1)$                                       `// Normalize to [0, 1] range.`

    `// Use ground truth and sample from prior to sample a noisy interpolant at timestep` $l$`.`

26     $\{\epsilon_{\text{coord}}, \epsilon_{\text{vis}}, \epsilon_{\text{conf}}\} \sim \{\mathcal{N}(0, I), \mathcal{N}(0, I), \mathcal{N}(0, I)\}$

27     $\tilde{P}_{n_T, l} \leftarrow l' * P_{n_T} + (1 - l') * \tilde{P}_{n_T, 1} + l' * \sigma_{\text{coord}}\epsilon_{\text{coord}}$

28     $\tilde{V}_{n_T, l} \leftarrow l' * V_{n_T} + (1 - l') * \tilde{V}_{n_T, 1} + l' * \sigma_{\text{vis}}\epsilon_{\text{vis}}$

29     $C_{n_T, 1, l} \leftarrow 1 - \min((\mathtt{stopgrad}(\tilde{P}_{n_T, l}) - P_{n_T})^2, 16^2)/16^2$           `// Ground truth confidence.`

30     $\tilde{C}_{n_T, l} \leftarrow l' * C_{n_T, 1, l} + (1 - l') * \tilde{C}_{n_T, 1} + l' * \sigma_{\text{conf}}\epsilon_{\text{conf}}$

    `// Set initial ground truth estimate to sample at timestep` $l$`.`

31     $\{\hat{P}_{n_T, 1, l}, \hat{V}_{n_T, 1, l}, \hat{C}_{n_T, 1, l}\} \leftarrow \{\tilde{P}_{n_T, l}, \tilde{V}_{n_T, l}, \tilde{C}_{n_T, l}\}$

32     **for** $k \leftarrow 1$ **to** $K - 1$ **do**

        `// Sample multiscale feature crops centred at current ground truth estimate.`

33       $f^{\Omega(\hat{P}_{n_T, k, l})} \leftarrow \mathtt{sample\_feats}(\{F^s_{n_T}\}_{s=1}^S, \Omega(\mathtt{cat}(\{\hat{P}_{n_T, k, l}, \mathtt{frame\_inds}\}, 3)))$

        `// Compute multiscale 4D correlation features.`

34       $\mathtt{Corr} \leftarrow \mathtt{cat}(\{\mathcal{M}_\rho((f^{\Omega(P^q)}_s)^\top (f^{\Omega(\hat{P}_{n_T, k, l})}_s))\}_{s=1}^S, 3) \in \mathbb{R}^{T \times N_{Pq} \times (S \times D_\mathcal{M})}$

        `// Prepare input and conditioning and compute refinement delta.`

35       $\mathtt{forward\_disp} \leftarrow \mathtt{cat}(\{0, \hat{P}_{n_T, k, l, 2:T} - \hat{P}_{n_T, k, l, 1:(T-1)}\}, 1) \in \mathbb{R}^{T \times N_{Pq} \times 2}$

36       $\mathtt{backward\_disp} \leftarrow \mathtt{cat}(\{\hat{P}_{n_T, k, l, 1:(T-1)} - \hat{P}_{n_T, k, l, 2:T}, 0\}, 1) \in \mathbb{R}^{T \times N_{Pq} \times 2}$

37       $\mathtt{disp} \leftarrow \mathtt{cat}(\{\mathtt{forward\_disp}, \mathtt{backward\_disp}\}, 3) \in \mathbb{R}^{T \times N_{Pq} \times 4}$

38       $\mathtt{input} \leftarrow \mathtt{cat}(\{\mathtt{disp}, \mathtt{sigmoid}(\hat{V}_{n_T, k, l}), \mathtt{sigmoid}(\hat{C}_{n_T, k, l}), l', \mathtt{prior\_flag}, \mathtt{frame\_inds}\}, 3)$

39       $\mathtt{conditioning} \leftarrow \mathtt{Corr}$

40       $\{\Delta\hat{P}_{n_T, k, l}, \Delta\hat{V}_{n_T, k, l}, \Delta\hat{C}_{n_T, k, l}\} \leftarrow \mathcal{T}_\theta(\mathtt{input}, \mathtt{conditioning})$

        `// Refine ground truth estimate.`

41       $\{\hat{P}_{n_T, (k+1), l}, \hat{V}_{n_T, (k+1), l}, \hat{C}_{n_T, (k+1), l}\} \leftarrow$
          $\{\hat{P}_{n_T, k, l} + \Delta\hat{P}_{n_T, k, l}, \hat{V}_{n_T, k, l} + \Delta\hat{V}_{n_T, k, l}, \hat{C}_{n_T, k, l} + \Delta\hat{C}_{n_T, k, l}\}$

        `// Compute losses.`

42       $\mathcal{L}_{\text{coord}} \leftarrow \frac{0.05}{N_T K}|P_{n_T} - \hat{P}_{n_T, (k+1), l}|$

43       $\mathcal{L}_{\text{vis}} \leftarrow \frac{1}{N_T K}\mathtt{BCE}(V_{n_T}, \mathtt{sigmoid}(\hat{V}_{n_T, (k+1), l}))$

44       $C_{n_T, (k+1), l} \leftarrow 1 - \min((\mathtt{stopgrad}(\hat{P}_{n_T, (k+1), l}) - P_{n_T})^2, 16^2)/16^2$   `// Ground truth confidence.`

45       $\mathcal{L}_{\text{conf}} \leftarrow \frac{1}{N_T K}|C_{n_T, (k+1), l} - \hat{C}_{n_T, (k+1), l}|$

46       $\mathcal{L} \leftarrow \mathcal{L} + \mathcal{L}_{\text{coord}} + \mathcal{L}_{\text{vis}} + \mathcal{L}_{\text{conf}}$

47     **end**

48   **end**

---

**Algorithm 2:** Sampling with GenPT

**Input**  : Transformer $\mathcal{T}_\theta$, encoder $\mathcal{F}_\phi$, MLP $\mathcal{M}_\rho$, tuple $\{I, P^q\}$.
**Output** : Samples $\{\tilde{P}, \tilde{V}, \tilde{C}\}$.
**Note**  : Assume tensor broadcasting when no explicit repeats are performed. $\mathtt{cat}(\{\cdot, \ldots\}, n)$ concatenates a tuple of tensors along
      the $n^{\text{th}}$ axis. $\mathtt{slice}(\mathtt{I},\ \mathtt{T},\ \mathtt{n})$ slices tensor $I$ into $N_T$ overlapping windows of size $T$ along its $n^{\text{th}}$ axis.

**1** $P^q \leftarrow \mathtt{repeat}(P^q, \text{``}N_{Pq}\ 3 \rightarrow T\ N_{Pq}\ 3\text{''}) \in \mathbb{R}^{T \times N_{Pq} \times 3}$

**2** $F \leftarrow \mathtt{get\_fmaps\_pyramid}(\mathcal{F}_\phi, I, S) \in \mathbb{R}^{S \times T' \times D_\mathcal{F} \times H/2^{s+1} \times W/2^{s+1}}$
  // Sample multiscale feature crops centred at query points.

**3** $f^{\Omega(P^q)} \leftarrow \mathtt{sample\_feats}(F, \Omega(P^q)) \in \mathbb{R}^{S \times T \times N_{Pq} \times D_\mathcal{F} \times 49}$
  // Slice into $N_T$ overlapping windows of size $T$.

**4** $F \leftarrow \mathtt{slice}(F, T, 2) \in \mathbb{R}^{N_T \times S \times T \times D_\mathcal{F} \times H/2^{s+1} \times W/2^{s+1}}$

**5** $\mathcal{L} \leftarrow 0$
  // Initialize ground truth estimates.

**6** $\{\hat{P}, \hat{V}, \hat{C}\} \leftarrow \{\mathbf{0} \in \mathbb{R}^{N_T \times K \times L \times T \times N_{Pq} \times 2}, \mathbf{0} \in \mathbb{R}^{N_T \times K \times L \times T \times N_{Pq} \times 1}, \mathbf{0} \in \mathbb{R}^{N_T \times K \times L \times T \times N_{Pq} \times 1}\}$
  // Initialize samples.

**7** $\{\tilde{P}, \tilde{V}, \tilde{C}\} \leftarrow \{\mathbf{0} \in \mathbb{R}^{N_T \times L \times T \times N_{Pq} \times 2}, \mathbf{0} \in \mathbb{R}^{N_T \times L \times T \times N_{Pq} \times 1}, \mathbf{0} \in \mathbb{R}^{N_T \times L \times T \times N_{Pq} \times 1}\}$
  // Iterate over windows.

**8** **for** $n_T \leftarrow 1$ **to** $N_T$ **do**

**9**  $\mathtt{frame\_inds} \leftarrow \mathtt{repeat}(\mathtt{cat}(\{t\}_{t=0}^{T-1}, 1), \text{``}T \rightarrow T\ N_{Pq}\ 1\text{''}) \in \mathbb{R}^{T \times N_{Pq} \times 1}$

**10**  $\{\epsilon_{\text{coord}}, \epsilon_{\text{vis}}, \epsilon_{\text{conf}}\} \sim \{\mathcal{N}(0, I), \mathcal{N}(0, I), \mathcal{N}(0, I)\}$

**11**  **if** $n_T = 1$ **then**
    // In first window. Sample from first prior.

**12**    $\{\tilde{P}_{n_T,1}, \tilde{V}_{n_T,1}, \tilde{C}_{n_T,1}\} \leftarrow \{P^q + \sigma_{\text{coord}}\epsilon_{\text{coord}},\ \sigma_{\text{vis}}\epsilon_{\text{vis}},\ \sigma_{\text{conf}}\epsilon_{\text{conf}}\}$

**13**    $\mathtt{prior\_flag} \leftarrow \mathbf{1} \in \mathbb{R}^{T \times N_{Pq} \times 1}$

**14**  **else**
    // Not in first window. Sample from second prior by using samples from previous
      window. Only add noise in the latter half of the window.

**15**    $\{\tilde{P}_{n_T,1,1:(T/2)}, \tilde{P}_{n_T,1,(T/2+1):T}\} \leftarrow \{\tilde{P}_{(n_T-1),L,(T/2+1):T},\ \tilde{P}_{(n_T-1),L,T} + \sigma_{\text{coord}}\epsilon_{\text{coord}}\}$

**16**    $\{\tilde{V}_{n_T,1,1:(T/2)}, \tilde{V}_{n_T,1,(T/2+1):T}\} \leftarrow \{\tilde{V}_{(n_T-1),L,(T/2+1):T},\ \tilde{V}_{(n_T-1),L,T} + \sigma_{\text{vis}}\epsilon_{\text{vis}}\}$

**17**    $\{\tilde{C}_{n_T,1,1:(T/2)}, \tilde{C}_{n_T,1,(T/2+1):T}\} \leftarrow \{\tilde{C}_{(n_T-1),L,(T/2+1):T},\ \tilde{C}_{(n_T-1),L,T} + \sigma_{\text{conf}}\epsilon_{\text{conf}}\}$

**18**    $\mathtt{prior\_flag} \leftarrow \mathbf{0} \in \mathbb{R}^{T \times N_{Pq} \times 1}$

**19**  **end**
  // Iterate over integration steps.

**20**  **for** $l \leftarrow 1$ **to** $L - 1$ **do**

**21**    $l' \leftarrow \mathtt{repeat}(l, \text{``}1 \rightarrow T\ N_{Pq}\ 1\text{''}) \in \mathbb{R}^{T \times N_{Pq} \times 1}$

**22**    $l' \leftarrow (l' - 1)/(L - 1)$               // Normalize to [0, 1] range.
    // Set initial ground truth estimate to sample at timestep $l$.

**23**    $\{\hat{P}_{n_T,1,l}, \hat{V}_{n_T,1,l}, \hat{C}_{n_T,1,l}\} \leftarrow \{\tilde{P}_{n_T,l}, \tilde{V}_{n_T,l}, \tilde{C}_{n_T,l}\}$
    // Iterate over refinement steps.

**24**    **for** $k \leftarrow 1$ **to** $K - 1$ **do**
      // Sample multiscale feature crops centred at current ground truth estimate.

**25**      $f^{\Omega(\hat{P}_{n_T,k,l})} \leftarrow \mathtt{sample\_feats}(F_{n_T}, \Omega(\mathtt{cat}(\{\hat{P}_{n_T,k,l}, \mathtt{frame\_inds}\}, 3)))$
      // Compute multiscale 4D correlation features.

**26**      $\mathtt{Corr} \leftarrow \mathtt{cat}(\{\mathcal{M}_\rho((f_s^{\Omega(P^q)})^\top (f_s^{\Omega(\hat{P}_{n_T,k,l})}))\}_{s=1}^S, 3) \in \mathbb{R}^{T \times N_{Pq} \times (S \times D_\mathcal{M})}$
      // Prepare input and conditioning and compute refinement delta.

**27**      $\mathtt{forward\_disp} \leftarrow \mathtt{cat}(\{0, \hat{P}_{n_T,k,l,2:T} - \hat{P}_{n_T,k,l,1:(T-1)}\}, 1) \in \mathbb{R}^{T \times N_{Pq} \times 2}$

**28**      $\mathtt{backward\_disp} \leftarrow \mathtt{cat}(\{\hat{P}_{n_T,k,l,1:(T-1)} - \hat{P}_{n_T,k,l,2:T}, 0\}, 1) \in \mathbb{R}^{T \times N_{Pq} \times 2}$

**29**      $\mathtt{disp} \leftarrow \mathtt{cat}(\{\mathtt{forward\_disp}, \mathtt{backward\_disp}\}, 3) \in \mathbb{R}^{T \times N_{Pq} \times 4}$

**30**      $\mathtt{input} \leftarrow \mathtt{cat}(\{\mathtt{disp}, \mathtt{sigmoid}(\hat{V}_{n_T,k,l}), \mathtt{sigmoid}(\hat{C}_{n_T,k,l}), l', \mathtt{prior\_flag}, \mathtt{frame\_inds}\}, 3) \in$
       $\mathbb{R}^{T \times N_{Pq} \times 9}$

**31**      $\mathtt{conditioning} \leftarrow \mathtt{Corr}$

**32**      $\{\Delta\hat{P}_{n_T,k,l}, \Delta\hat{V}_{n_T,k,l}, \Delta\hat{C}_{n_T,k,l}\} \leftarrow \mathcal{T}_\theta(\mathtt{input}, \mathtt{conditioning})$
      // Refine ground truth estimate.

**33**      $\{\hat{P}_{n_T,(k+1),l}, \hat{V}_{n_T,(k+1),l}, \hat{C}_{n_T,(k+1),l}\} \leftarrow$
       $\{\hat{P}_{n_T,k,l} + \Delta\hat{P}_{n_T,k,l}, \hat{V}_{n_T,k,l} + \Delta\hat{V}_{n_T,k,l}, \hat{C}_{n_T,k,l} + \Delta\hat{C}_{n_T,k,l}\}$

**34**    **end**
    // Compute vector field using ground truth estimate and sample from prior.

**35**    $\{\tilde{P}^{\text{flow}}, \tilde{V}^{\text{flow}}, \tilde{C}^{\text{flow}}\} \leftarrow \{\hat{P}_{n_T,K,l} - \tilde{P}_{n_T,1}, \hat{V}_{n_T,K,l} - \tilde{V}_{n_T,1}, \hat{C}_{n_T,K,l} - \tilde{C}_{n_T,1}\}$
    // Integrate to next sample using vector field estimate.

**36**    $l'' \leftarrow \mathtt{repeat}(l + 1, \text{``}1 \rightarrow T\ N_{Pq}\ 1\text{''}) \in \mathbb{R}^{T \times N_{Pq} \times 1}$

**37**    $l'' \leftarrow (l'' - 1)/(L - 1)$

**38**    $\Delta l' \leftarrow l'' - l'$

**39**    $\{\tilde{P}_{n_T,(l+1)}, \tilde{V}_{n_T,(l+1)}, \tilde{C}_{n_T,(l+1)}\} \leftarrow \{\tilde{P}_{n_T,l} + \Delta l' \tilde{P}^{\text{flow}}, \tilde{V}_{n_T,l} + \Delta l' \tilde{V}^{\text{flow}}, \tilde{C}_{n_T,l} + \Delta l' \tilde{C}^{\text{flow}}\}$

**40**  **end**

**41** **end**

