# OpenReview forum: "Generative Point Tracking with Flow Matching"
_ICLR.cc/2026/Conference — Submitted to ICLR 2026_

### Official Review · Reviewer_837U · 2025-10-27

**Soundness:** 1
**Presentation:** 2
**Contribution:** 1
**Rating:** 2
**Confidence:** 5

**Summary:**

This paper introduces Generative Point Tracker (GenPT), which reinterprets the iterative optimization paradigm of many of the modern point trackers (such as PIPs, CoTracker3, LocoTrack) as a form of flow matching. The authors claim to bridge point tracking and generative modeling by formulating correspondence estimation as learning a continuous denoising process that maps perturbed query coordinates to target positions. The framework introduces Gaussian perturbations to query points, defines an auxiliary velocity field trained with a flow-matching objective, and evaluates both single-sample and Best-of-N inference strategies. The paper also includes a multi-template tracking extension, performing patch-wise correspondence aggregation inspired by LocoTrack.

**Strengths:**

- The paper tackles a genuine limitation of current discriminative point trackers, their inability to represent uncertainty and multimodal hypotheses in ambiguous or occluded regions.
- The authors provide comprehensive comparisons across several datasets

**Weaknesses:**

### Lack of generative insight
Although the paper positions itself as a generative reformulation of tracking, the actual mechanism remains deterministic iterative optimization under Gaussian perturbation, not a generative process.
- In generative models (diffusion or rectified flow), the model learns to map **pure noise --> data samples**, learning meaningful dynamics along a linear trajectory in data space.
- In GenPT, the model learns **query + noise --> correspondence**, where the starting point already encodes the spatial identity of the tracked feature. The added noise does not represent a generative latent, only a small random offset to an already meaningful input.
- Thus, the flow is effectively a regularized refinement of supervised training, not a learned stochastic trajectory from noise to data.
- Equation 6 changes the standard CoTracker initialization when $l=0$; increasing $l$ simply reduces supervision strength, not adding new semantics.
- In essence, GenPT = CoTracker3 + Gaussian perturbation + renaming of loss, rather than a true flow-matching model.

### Evaluation issues
- The Best-of-N performance gains could stem entirely from multiple inference-time noise injections, not a learned generative diversity. No comparison to a simple CoTracker3 + random perturbation at inference baseline is provided.
- The empirical improvements are small and inconsistent, and the method fails to demonstrate meaningful benefits in standard single-sample evaluation.

### Presentation and clarity
- The notation is excessive, making the method difficult to follow.

### Overall
While the paper explores a creative framing of point tracking via flow matching, it does not deliver genuine generative insight or methodological novelty. The proposed approach is functionally equivalent to noisy supervised fine-tuning of existing trackers, with only minor differences in objective formulation. The results and framing overstate the impact relative to the simplicity of the actual change.

**Questions:**

See weaknesses

---

> ### Author Response · Authors · 2025-11-19
> **Initial response part 1**
>
> > **In generative models (diffusion or rectified flow), the model learns to map pure noise --> data samples, learning meaningful dynamics along a linear trajectory in data space.**
>
> While it is true that for most diffusion models, the model can only be trained to map standard Gaussian noise to data samples, this is not the case for flow matching models. Unlike diffusion models, there is no restriction on the source distribution being standard Gaussian noise [1, 2, 3]. This is stated in L153-155 in our manuscript. The source distribution can be a non-standard Gaussian (e.g., centred at query points), or some other distribution (e.g., another data distribution). This is a unique feature of flow matching that is, understandably, not so well known, but has been exploited for text-to-image generation (and vice-versa) [2], image-to-depth generation [2], image super-resolution [2, 3], and image in-painting [3].
>
> [1] Alexander Tong, Kilian Fatras, Nikolay Malkin, Guillaume Huguet, Yanlei Zhang, Jarrid Rector-Brooks, Guy Wolf, and Yoshua Bengio. Improving and generalizing flow-based generative models with minibatch optimal transport. TMLR, 2024.
>
> [2] Qihao Liu, Xi Yin, Alan Yuille, Andrew Brown, and Mannat Singh. Flowing from Words to Pixels: A Noise-Free Framework for Cross-Modality Evolution. In CVPR, pp. 2755–2765, 2025.
>
> [3] Michael S Albergo, Mark Goldstein, Nicholas M Boffi, Rajesh Ranganath, and Eric Vanden-Eijnden. Stochastic interpolants with data-dependent couplings. ICML, 2024.
>
> > **In GenPT, the model learns query + noise --> correspondence, where the starting point already encodes the spatial identity of the tracked feature. The added noise does not represent a generative latent, only a small random offset to an already meaningful input.**
>
> We strongly disagree: GenPT learns the conditional velocity field between a conditional prior and a conditional data distribution. The velocity field induces a conditional probability path between the two distributions.
>
> In our case, the conditional prior is window-dependent, where in the first window it is a Gaussian centred about query points for point trajectories and is a standard Gaussian for visibilities and confidences. In the second window, it is a Gaussian centred about estimates made in the previous window.
>
> Flow matching allows us to use such a non-standard prior, which imposes a bias that helps with tracking accuracy. This bootstrapped prior is still stochastic and contains only partial information related to the conditional data distribution (a form of "data-dependent coupling" [2, 3]).
>
> The conditional data distribution is a sum of Dirac delta functions, each centred about ground truth point trajectories, visibilities, and confidences, given the video and query points.
>
> The conditional probability path between the conditional prior distributions and conditional data distribution is a Gaussian whose mean is defined as the linear interpolation between a sample from the two distributions, with a tuned variance schedule. This path is conditioned on the query points, video, ground truth sample, and prior sample. Once learned, GenPT can sample from the data distribution through ODE integration, starting from the prior and guided by its approximated vector field.
>
> > **Equation 6 changes the standard CoTracker initialization when l = 0; increasing l  simply reduces supervision strength, not adding new semantics.**
>
> Equation 6 defines the conditional probability path between the conditional prior and data distributions. Such a path is necessary in training a flow matching model [1], where the model is trained to map a sample anywhere along this path to a sample from the data distribution. Indeed, when l’ = 0, the conditional sample is from the prior, which is fashioned similarly to CoTracker3’s initialization but with additional noise (when in the first window). When l’ approaches 1, the conditional sample contains more information of the ground truth. This training dynamic is common in all diffusion models, i.e., during training, depending on what timestep was sampled (l’ in our case), a diffusion model can be tasked to denoise a very noisy input (the ground truth with a lot of noise) or a clean input (the ground truth with no noise).
>
> > **In essence, GenPT = CoTracker3 + Gaussian perturbation + renaming of loss, rather than a true flow-matching model.**
>
> CoTracker3 does not model the conditional probability path described above (or any probability path) and adding noise to its initialization does not change this. In contrast, GenPT is explicitly trained to model the conditional probability path in Equation 6, as required by flow matching methods [1]. Its prior is purpose-built to exploit partial data (query points or previous estimates), a capability uniquely enabled by flow matching [2, 3].

---

> ### Author Response · Authors · 2025-11-19
> **Initial response part 2**
>
> > **The Best-of-N performance gains could stem entirely from multiple inference-time noise injections, not a learned generative diversity. No comparison to a simple CoTracker3 + random perturbation at inference baseline is provided.**
>
> As suggested by the reviewer, we include a new comparison between GenPT and CoTracker3 with test-time random perturbations in the appendix (A.14). Tables 14-16 show our quantitative results. Tables 11-13 show results for a similar comparison, but between GenPT and its discriminatively-trained variant (which we’ll call GenPT-D here).
>
> In a single shot (when N = 1), GenPT outperforms CoTracker3 in all tested benchmarks (PointOdyssey, Dynamic Replica, a subset of both TAP-Vid and TAP-Vid Sliding Occluder) on both visible and occluded point tracking accuracy, with more than a 20% percentage difference in some cases. The same applies to GenPT vs GenPT-D.
>
> To show that GenPT generates trajectories that lie closer to the true data distribution compared to its discriminatively-trained competitor, we observe both models’ peak performance (best of N), their worst performance (worst of N), and the change in the difference between their best of N and worst of N tracking accuracy as N increases (as a rough measure of variance), with a focus on occluded point tracking accuracy.
>
> In summary, GenPT consistently has the highest tracking accuracy for both best of N and worst of N, while experiencing the lowest reduction in tracking accuracy as N increases in worst of N, especially on occluded points. In other words, in areas of uncertainty, GenPT is capable of capturing meaningful modes, unlike its discriminatively-trained counterparts with test-time random perturbations.
>
> In our initial submission, we also provided qualitative evidence in Figures 6 and 7 in the appendix and in the "discriminative vs generative" folder in the supplemental videos.
>
> > **The empirical improvements are small and inconsistent, and the method fails to demonstrate meaningful benefits in standard single-sample evaluation.**
>
> In the single-shot setting (no best-of-N), GenPT operates at 2x the speed of CoTracker3 while outperforming its visible and occluded point tracking accuracy on all test sets when both models are trained in an identical training setup.
>
> When comparing against the best pre-trained CoTracker3 model (Kub+15k, which is more data than what we trained GenPT with), our best GenPT model (POD) only underperforms CoTracker3 (Kub+15k) on visible points in the TAP-Vid benchmark (by 1.4%). However, GenPT outperforms it on both visible and occluded points in the PointOdyssey (by 0.9%-vis and 2.7%-occ), Dynamic Replica (by 1.8%-vis and 5.4%-occ), and TAP-Vid Sliding Occluder (by 0.1%-vis and 10.6%-occ) benchmarks. We believe this to be a consistent improvement over the baseline, with a significant improvement on occluded point tracking accuracy, showcasing GenPT’s efficiency, and the benefits of the generative casting of the task.

---

> ### Author Response · Authors · 2025-11-19
> **Initial response part 3**
>
> > **Overall: While the paper explores a creative framing of point tracking via flow matching, it does not deliver genuine generative insight or methodological novelty. The proposed approach is functionally equivalent to noisy supervised fine-tuning of existing trackers, with only minor differences in objective formulation. The results and framing overstate the impact relative to the simplicity of the actual change.**
>
> We thank the reviewer for acknowledging the creative framing of our approach. To summarize, we introduced the following non-trivial modifications to vanilla flow matching + point tracking and demonstrate its efficacy empirically:
>
> 1. A double loop where we have the typical ODE integration loop on the outside and the ground truth estimation loop on the inside. The inner loop was included during training, meaning that at each training step, we had to perform an iterative refinement to arrive at an estimate of the ground truth. This is in contrast to vanilla flow matching, which only contains a single loop (for ODE integration) and is trained with no iterative refinement. This is also in contrast to CoTracker3 and other point trackers, which are not trained to model a  conditional probability path. We are marrying the two worlds, essentially.
> 2. We used a window-dependent prior to link samples across windows. Although conceptually it's similar to what CoTracker3 and other point trackers do (minus the additive noise), it is not how one would typically model their prior when applying flow matching, which would usually be a standard Gaussian. We took advantage of a rarely-used feature of flow matching [1, 2, 3] where the prior (i.e., source distribution) can be any parameterized distribution.
> 3. We used a specialized variance schedule in the conditional probability path that is tuned for point trajectories.
>
> All three changes were ablated in the appendix (A.5, A.6, and A.7) of the initial submission and each were shown to have a substantial effect on tracking performance. These modifications were necessary to have both a valid flow matching formulation and an effective point tracker.

---

> ### Comment · Reviewer_837U · 2025-11-26
>
> Thank you for the detailed rebuttal. I have carefully reviewed the authors’ responses and the other reviews.
> * **Clarification on flow matching source distributions.** The authors correctly note that flow matching does not require a standard Gaussian source distribution. However, this does not address the underlying concern. The question is not whether such a prior is allowed, but whether the proposed formulation introduces meaningful stochasticity or generative behavior for point tracking.
> * **Conditional velocity field and probability path.** The rebuttal argues that GenPT models a conditional velocity field and that iterative refinement can be interpreted as a conditional FM ODE. While this interpretation is mathematically valid, it does not respond to the main question: does the model exhibit generative behavior beyond what a deterministic tracker with noise injection can already produce? The current prior (query location plus Gaussian perturbation) is not a meaningful latent variable, but a small offset around an already informative initialization, consistent with standard iterative update schemes. This setup does not yield multimodal hypotheses or uncertainty modeling beyond what CoTracker3 can approximate with direct perturbations. The effective dynamics remain governed by deterministic refinement rather than learned ODE integration, and the model does not synthesize new correspondences but reallocates refinement trajectories based on the same underlying structure.
> The discussion of Eq. 6 also conflates training-time interpolation with inference-time behavior. My critique concerns how inference operates, not the interpolation schedule used during training.
> * **Suitability of point tracking for generative modeling.** As noted by Reviewer QTUL, point tracking may not naturally benefit from a generative formulation. Much of GenPT’s behavior closely mirrors CoTracker3’s deterministic iterative updates. The rebuttal does not convincingly demonstrate that modeling a flow path between the proposed “prior” and the data distribution provides new modeling capacity or addresses any open challenge in point tracking. The generative framing appears to be an interpretation rather than a source of demonstrable benefits.
> * **Overall.** These issues appear from a **fundamental mismatch between generative modeling and the deterministic nature of point tracking.** GenPT ultimately relies on the same mechanisms that make CoTracker3 effective, and I remain unconvinced that the task requires, or benefits from, its flow-matching reinterpretation.

---

> > ### Author Response · Authors · 2025-12-04
> > **Second rebuttal response part 1**
> >
> > > **[From Reviewer 837U's initial review] In generative models (diffusion or rectified flow), the model learns to map pure noise --> data samples, learning meaningful dynamics along a linear trajectory in data space.**
> >
> > > **[From Reviewer 837U's rebuttal response: point 1] Clarification on flow matching source distributions. The authors correctly note that flow matching does not require a standard Gaussian source distribution.**
> >
> > Reviewer 837U now acknowledges that flow matching-based generative models do not require a Gaussian prior. **GenPT is a conditional generative model** that uses a flow matching-based formulation with a non-standard-Gaussian conditional prior, consistent with flow matching models that use a data-dependent coupling [1, 2].
> >
> > > **[From Reviewer 837U's initial review] The Best-of-N performance gains could stem entirely from multiple inference-time noise injections, not a learned generative diversity. No comparison to a simple CoTracker3 + random perturbation at inference baseline is provided.**
> >
> > > **[From Reviewer 837U's rebuttal response: point 1] The question is not whether such a prior is allowed, but whether the proposed formulation introduces meaningful stochasticity or generative behavior for point tracking.**
> >
> > > **[From Reviewer 837U's rebuttal response: point 1] I remain unconvinced that the task requires, or benefits from, its flow-matching reinterpretation.**
> >
> > > **[From Reviewer 837U's rebuttal response: point 2] This setup does not yield multimodal hypotheses or uncertainty modeling beyond what CoTracker3 can approximate with direct perturbations.**
> >
> > We have demonstrated, both empirically and theoretically, the benefits that such a generative formulation has on point tracking, particularly in cases of uncertainty, i.e., during occlusions. As requested by Reviewer 837U, we compared GenPT with CoTracker3 with test-time random perturbations and showed a drastic difference in occluded point tracking performance, with **more than a 20% difference in occluded point tracking performance** in some benchmarks. For example, in best-of-10 (guided by oracle) GenPT has 59.2% on d_occ_avg in Dynamic Replica while CoTracker3 has 38.7%. In a single shot, GenPT and CoTracker3 have 53.8% and 31.8% on d_occ_avg in Dynamic Replica, respectively. These results can be seen in Table 14. The rest of the results can be seen in Tables 14-16 in A.14 in the appendix.
> >
> > We also included additional comparisons (Tables 11-13) with a deterministic variant of GenPT with test-time random perturbations and showed similar differences in tracking performance. These experiments, along with the comparison with CoTracker3 with test-time random perturbations, were provided as part of our initial rebuttal, and it appears that Reviewer 837U did not acknowledge these experiments and their results.
> >
> > In summary, **GenPT generates trajectories that lie closer to the true data distribution compared to its discriminatively-trained competitors**, consistently having the highest accuracy in its best setting (best of N) and worst setting (worst of N), while experiencing the lowest reduction in tracking accuracy as N increases in worst of N, especially on occluded points. In other words, in areas of uncertainty, GenPT is capable of capturing meaningful modes, unlike its discriminatively-trained counterparts with test-time random perturbations. In addition to the quantitative evidence stated above, qualitative evidence can be found in Figures 6 and 7 in the appendix and in the "discriminative vs generative" folder in the supplemental videos.

---

> > > ### Author Response · Authors · 2025-12-04
> > > **Second rebuttal response part 2**
> > >
> > > > **[From Reviewer 837U's rebuttal response: point 3] As noted by Reviewer QTUL, point tracking may not naturally benefit from a generative formulation.**
> > >
> > > > **[From Reviewer QTUL's review] Point tracking is inherently a deterministic problem, so a multi-modal approach may not be well-suited for this task.**
> > >
> > > We disagree with this subjective statement from both Reviewer 837U and QTUL. **Point tracking is not inherently deterministic.** As we have stated in our response to Reviewer QTUL and in our initial submission, there are myriad cases of uncertainty inherently present in point tracking, such as occlusions, lighting changes, and sensor noise. To state that point tracking is a deterministic problem is to discount the numerous generative formulations of object tracking [3, 4, 5], optical flow estimation [6, 7, 8], and other related problems that are now emerging, and to ignore the inherent uncertainty in the tracking task.
> > >
> > >
> > > > **[From Reviewer 837U's rebuttal response: point 4] GenPT ultimately relies on the same mechanisms that make CoTracker3 effective, and I remain unconvinced that the task requires, or benefits from, its flow-matching reinterpretation.**
> > >
> > > **We disagree, GenPT is not a reinterpretation of CoTracker3.** While the two models both use correlations to help guide estimates, their implementations significantly diverge, reflecting their generative vs. discriminative modelling goals.
> > >
> > > As stated in our initial response, GenPT is explicitly trained to model the conditional probability path in Equation 6, as required by flow matching methods [9]. CoTracker3 does not model the conditional probability path described above (or any probability path) and adding noise to its initialization does not change this. In other words, **there is no probability flow ODE being solved in CoTracker3**, unlike in GenPT.
> > >
> > > Implementation-wise, this results in **GenPT consisting of two (nested) loops**: an outer loop integrating the ODE to generate the sample at the next step in the ODE path, and an inner loop for providing the ground truth estimate at the current step, which is used to take the next integration step. In contrast, **CoTracker3 consists of a single loop** and it’s just for the ground truth estimate, with no model of a probability path.
> > >
> > > FlowDiffuser [7] is the most similar to GenPT in this respect, as it also uses two loops, but its task is optical flow generation instead of point tracking, and they employ a more conventional diffusion model. Specifically, there is no flow matching, no sliding window processing, no window-dependent prior, no tuned variance schedule, and the task is to estimate the flow of pixels from one frame to the next instead of tracking a given set of pixels throughout an entire video. Importantly, their motivation for generative modelling of optical flow is the same as ours: to capture the uncertainty inherent in motion analysis—which discriminative models can not do—and thereby improve performance in these uncertain scenarios. GenPT is fundamentally different to CoTracker3 in much the same way that FlowDiffuser is fundamentally different to RAFT [10]: both are trying to analyze motion, both use iterative refinement guided by correlations, but the former is a generative model consisting of two loops (outer for probability path / sampling, inner for ground truth estimation) while the latter is a discriminative model consisting of one (for ground truth estimation). We will update the related work section in the paper to clarify these points.
> > >
> > > While the predominant work on point tracking has focused on discriminative methods, here we explore a generative modelling approach and demonstrate its benefits. Specifically, through our experiments comparing GenPT's occluded point tracking performance with discriminative variants' (Tables 11-16), **we have provided convincing evidence that point tracking benefits from GenPT's generative formulation, in some cases showing a more than 20% improvement in occluded point tracking accuracy**.

---

> > > > ### Author Response · Authors · 2025-12-04
> > > > **Second rebuttal response part 3 (references)**
> > > >
> > > > ## References
> > > > [1] Qihao Liu, Xi Yin, Alan Yuille, Andrew Brown, and Mannat Singh. Flowing from Words to Pixels: A Noise-Free Framework for Cross-Modality Evolution. In CVPR, pp. 2755–2765, 2025.
> > > >
> > > > [2] Michael S Albergo, Mark Goldstein, Nicholas M Boffi, Rajesh Ranganath, and Eric Vanden-Eijnden. Stochastic interpolants with data-dependent couplings. ICML, 2024.
> > > >
> > > > [3] Run Luo, Zikai Song, Lintao Ma, Jinlin Wei, Wei Yang, and Min Yang. DiffusionTrack: Diffusion Model For Multi-Object Tracking. In AAAI, 2024.
> > > >
> > > > [4] Fei Xie, Zhongdao Wang, Chao Ma. DiffusionTrack: Point Set Diffusion Model for Visual Object Tracking. In CVPR, 2024.
> > > >
> > > > [5] Weiyi Lv, Yuhang Huang, Ning Zhang, Ruei-Sung Lin, Mei Han, Dan Zeng. DiffMOT: A Real-time Diffusion-based Multiple Object Tracker with Non-linear Prediction. In CVPR, 2024.
> > > >
> > > > [6] Saurabh Saxena, Charles Herrmann, Junhwa Hur, Abhishek Kar, Mohammad Norouzi, Deqing Sun, and David J Fleet. The surprising effectiveness of diffusion models for optical flow and monocular depth estimation. In NeurIPS, volume 36, 2024.
> > > >
> > > > [7] Ao Luo, Xin Li, Fan Yang, Jiangyu Liu, Haoqiang Fan, and Shuaicheng Liu. FlowDiffuser: Advancing optical flow estimation with diffusion models. In CVPR, pp. 19167–19176, June 2024.
> > > >
> > > > [8] Christoph Strecha, Rik Fransens, and Luc Van Gool. A Probabilistic Approach to Large Displacement Optical Flow and Occlusion Detection. In Statistical Methods in Video Processing, pp. 71–82, 2004.
> > > >
> > > > [9] Alexander Tong, Kilian Fatras, Nikolay Malkin, Guillaume Huguet, Yanlei Zhang, Jarrid Rector-Brooks, Guy Wolf, and Yoshua Bengio. Improving and generalizing flow-based generative models with minibatch optimal transport. TMLR, 2024.
> > > >
> > > > [10] Zachary Teed, Jia Deng. RAFT: Recurrent All-Pairs Field Transforms for Optical Flow. In ECCV, 2020.

---

### Official Review · Reviewer_QTUL · 2025-11-01

**Soundness:** 3
**Presentation:** 3
**Contribution:** 2
**Rating:** 4
**Confidence:** 4

**Summary:**

This paper introduces GenPT, a generative point tracker that models multi-modal trajectories for long-range point tracking in videos. Unlike discriminative trackers that regress a single mean and thus struggle under occlusions or appearance changes, GenPT trains a likelihood-based model with flow matching and three key modifications: (i) iterative refinement within each step, (ii) a window-dependent prior, and (iii) a variance schedule tailored to point coordinates. GenPT achieves competitive visible-point accuracy and state-of-the-art occluded-point accuracy.

**Strengths:**

1. This paper introduces the first generative point tracker trained using a modified flow-matching objective for trajectories, extending generative modeling concepts to the task of point tracking.
2. The authors design three key modules: iterative refinement, window-dependent prior, and variance schedule. These components are well-motivated and thoroughly ablated.

**Weaknesses:**

1. Point tracking is inherently a deterministic problem, so a multi-modal approach may not be well-suited for this task.
2. The improvements of this model mainly target occluded points. However, the objective function used in models such as CoTracker3 or other similar approaches is typically L=Huber_loss(predicted point,ground truth point)×is_visible_gt(this point)
In other words, these models are not explicitly designed to predict occluded points.
3. The greedy search strategy requires running the algorithm five times, which makes it computationally expensive and time-consuming.

**Questions:**

1. Could you train CoTracker3 with the objective L=Huber_loss(predicted point,ground truth point) and evaluate how much improvement it achieves on occluded points?
2. Do the failure cases tend to cluster around homogeneous textures or repetitive patterns?

---

> ### Author Response · Authors · 2025-11-19
> **Initial response part. 1**
>
> > **Point tracking is inherently a deterministic problem, so a multi-modal approach may not be well-suited for this task.**
>
> We strongly disagree. While previous incarnations have addressed the task via deterministic means, generative modeling provides a more general framework to encompass uncertainty, which is inherent to the task. In particular, in real world settings there are many factors that introduce ambiguity when tracking a point, such as when the point is occluded or when it changes appearance (e.g., sensor noise or lighting change). We state this in the abstract, L40-44, and L102-109 in the main manuscript of our initial submission.
>
> These circumstances permit many possible solutions to the location of the point in a manner that’s not dissimilar to optical flow estimation and depth estimation. These ill-posed problems—which have traditionally been addressed via deterministic approaches—have recently been cast within generative formulations [1, 2] and other probabilistic formulations [3].
>
> Deterministic models are well known to regress to the mean solution (if trained with an L2 loss) or the mode solution (if trained with an L1 loss), which can result in a poor estimate that does not align with the underlying ground truth data distribution when there is uncertainty (i.e., many modes). With a generative model, you can sample from one of the many plausible modes during these cases of uncertainty, which results in an estimate that more accurately reflects the underlying ground truth distribution.
>
> In short, when tracked points are occluded, modelling multiple possible trajectories (as a multi-modal distribution) allows the model to make better decisions about how the trajectories should continue when the point reappears.
>
> [1] Saurabh Saxena, Charles Herrmann, Junhwa Hur, Abhishek Kar, Mohammad Norouzi, Deqing Sun, and David J Fleet. The surprising effectiveness of diffusion models for optical flow and monocular depth estimation. In NeurIPS, volume 36, 2024.
>
> [2] Ao Luo, Xin Li, Fan Yang, Jiangyu Liu, Haoqiang Fan, and Shuaicheng Liu. FlowDiffuser: Advancing optical flow estimation with diffusion models. In CVPR, pp. 19167–19176, June 2024.
>
> [3] Christoph Strecha, Rik Fransens, and Luc Van Gool. A Probabilistic Approach to Large Displacement Optical Flow and Occlusion Detection. In Statistical Methods in Video Processing, pp. 71–82, 2004.
>
>
> > **The improvements of this model mainly target occluded points. However, the objective function used in models such as CoTracker3 or other similar approaches is typically L=Huber_loss(predicted point,ground truth point)×is_visible_gt(this point) In other words, these models are not explicitly designed to predict occluded points.**
> >
> > **Q1. Could you train CoTracker3 with the objective L=Huber_loss(predicted point,ground truth point) and evaluate how much improvement it achieves on occluded points?**
>
> This summary of the CoTracker3 loss is not accurate. CoTracker3 is supervised on occluded points but its loss function (Equation 2 in [4]) downweights them to prioritize visible points. To provide a fair comparison, the CoTracker3 models we trained from scratch used identical loss weightings and training data setups as GenPT. In these controlled experiments, GenPT consistently outperforms CoTracker3 on both visible and occluded points (Tables 1 and 2). Furthermore, GenPT’s superior occluded point tracking accuracy is demonstrated even when comparing our Kub-trained GenPT to the pre-trained CoTracker3 models, which were trained on significantly more data (up to 10.4x more).
>
> [4] Nikita Karaev, Iurii Makarov, Jianyuan Wang, Natalia Neverova, Andrea Vedaldi, and Christian Rupprecht. CoTracker3: Simpler and better point tracking by pseudo-labelling real videos. arXiv preprint arXiv:2410.11831, 2024a.
>
> > **The greedy search strategy requires running the algorithm five times, which makes it computationally expensive and time-consuming.**
>
> As the first paper on generative point tracking, we believe our initial implementation of best-of-N to be a reasonable starting point for exploring ways to exploit the generative capabilities of GenPT beyond the basics (i.e., improved occluded point tracking accuracy and multi-modal modelling). Furthermore, GenPT already exhibits superior accuracy in the single-shot setting when compared to popular baselines, while being 2x faster than CoTracker3. As stated in the runtime analysis section in the appendix (A.16) of our initial submission, we believe there is room for improving the best-of-N setup, such as introducing adaptive sampling strategies (as Reviewer DBce pointed out).

---

> > ### Author Response · Authors · 2025-11-19
> > **Initial response part 2**
> >
> > > **Q2. Do the failure cases tend to cluster around homogeneous textures or repetitive patterns?**
> >
> > Yes, failure cases tend to cluster around homogeneous textures or repetitive patterns. However, we find this to be much more frequent when the tracked point is occluded. All point trackers we’ve tested rarely fail when the point is visible, even when the queried point is on a region devoid of unique features (e.g., homogeneous textures). This is due to their spatial context and temporal prior.
> >
> > As shown in prior work [4, 5], failures tend to occur when the point is occluded, as evidenced by reduced tracking accuracy when the point is occluded compared to when it’s visible. What is not well known—and what we have revealed—is that failures are even more pronounced for discriminative models when the tracked point is both occluded and is tracking a feature that is not uniquely identifiable (e.g., a point on a homogeneously-textured object). Here we see a much clearer difference between discriminative models and GenPT. In cases of uncertainty, discriminative models collapse to a single, often incorrect mean/mode estimate, which, in the case described previously, is exacerbated by false positives from the correlation mechanism. In contrast, GenPT’s generative modelling predicts a more plausible location by leveraging a strengthened understanding of dynamics.
> >
> > This difference can be quantitatively observed on the RGB-S test set of the TAP-Vid Sliding Occluder benchmark (Table 2), where GenPT more easily overcomes false positives from the correlation mechanism caused by texture homogeneity. It can also be qualitatively observed in the video with pigs in the DAVIS test set of the TAP-Vid benchmark, under the “discriminative vs generative” folder in the supplemental videos of the initial submission. In the case of the pig example, the tracked point is a point on the pig, which is not uniquely identifiable. As soon as the pig turns around (self-occlusion), GenPT continues to successfully track the point while its discriminative variant fails. Since the majority of points in standard benchmarks are visible, this particular failure case has been rarely investigated.
> >
> > [5] Nikita Karaev, Ignacio Rocco, Benjamin Graham, Natalia Neverova, Andrea Vedaldi, and Christian Rupprecht. CoTracker: It is better to track together. In ECCV, pp. 18–35, 2024b.

---

### Official Review · Reviewer_DBce · 2025-11-03

**Soundness:** 3
**Presentation:** 3
**Contribution:** 3
**Rating:** 6
**Confidence:** 4

**Summary:**

The paper introduces the Generative Point Tracker (GenPT), the first framework to address the Point Tracking problem using a Generative Model based on Flow Matching. Existing Discriminative Models struggle with uncertainty (e.g., occlusion) as they regress to a single mean estimate. GenPT overcomes this by modeling multi-modal trajectories, enabling it to sample several plausible paths in ambiguous situations.

**Strengths:**

- GenPT can model and sample from multiple plausible trajectory candidates, particularly when tracking uncertainty is high due to occlusion. This translates directly to state-of-the-art tracking accuracy on occluded points.
- The model effectively transitions between probabilistic and quasi-deterministic behavior. While always generative, its prediction variance tightly contracts (becoming nearly deterministic) when the tracked point is clearly visible and uniquely identifiable.

**Weaknesses:**

- There is a substantial and recurring performance gap between the Oracle scores (the model's maximum potential) and the Greedy scores (the model's actual performance when relying on its confidence). This fundamental disconnect means the model is poor at judging the quality of the trajectories it generates, limiting the real-world utility of its multi-modality.
- The advertised speed advantage (2x faster than CoTracker3) is strictly limited to generating a single sample. To achieve the demonstrated improvements in accuracy, the 'Best-of-N' sampling method must be used. This process rapidly increases the runtime, often making GenPT slower than its discriminative counterparts, thus sacrificing one of its key efficiency claims for practical performance.
- A significant portion of GenPT's SOTA claim relies on the custom TAP-Vid Sliding Occluder Benchmark introduced by the authors, which is specifically designed to highlight its strength in occlusion handling. While useful, the novelty of the benchmark means the competitive results require independent verification across established, universally adopted benchmarks.

**Questions:**

Have the authors explored an adaptive sampling strategy where multiple samples ('Best-of-N') are only generated in windows where the model's initial predicted uncertainty (variance/confidence) is above a certain threshold, rather than sampling N times in every window?

---

> ### Author Response · Authors · 2025-11-19
>
> > **There is a substantial and recurring performance gap between the Oracle scores (the model's maximum potential) and the Greedy scores (the model's actual performance when relying on its confidence). This fundamental disconnect means the model is poor at judging the quality of the trajectories it generates, limiting the real-world utility of its multi-modality.**
>
> While the disconnect between the oracle scores and the greedy scores shows that there’s room for improvement of the model’s ability to estimate its own confidence, we believe that GenPT still serves as a solid starting point for generative modelling of point trajectories. Ignoring the best-of-N setting, GenPT still has superior point tracking performance, especially on occluded points, when compared to its discriminative counterpart (see A.9 in the appendix and Table 7) and competing baselines. We believe the improvements to occluded point tracking accuracy alone to have real-world utility of its generative modelling.
>
> > **The advertised speed advantage (2x faster than CoTracker3) is strictly limited to generating a single sample. To achieve the demonstrated improvements in accuracy, the 'Best-of-N' sampling method must be used. This process rapidly increases the runtime, often making GenPT slower than its discriminative counterparts, thus sacrificing one of its key efficiency claims for practical performance.**
>
> In the single-shot setting (no best-of-N), GenPT operates at 2x the speed of CoTracker3 while outperforming its visible and occluded point tracking accuracy on all test sets when both models are trained in an identical training setup (see Table 1 and 2).
>
> When comparing against the best pre-trained CoTracker3 model (Kub+15k, which is more data than what we trained GenPT with), our best GenPT model (POD) only underperforms CoTracker3 (Kub+15k) on visible points in the TAP-Vid benchmark (by 1.4%). However, GenPT outperforms it on both visible and occluded points in the PointOdyssey (by 0.9%-vis and 2.7%-occ), Dynamic Replica (by 1.8%-vis and 5.4%-occ), and TAP-Vid Sliding Occluder (by 0.1%-vis and 10.6%-occ) benchmarks. We believe this to be a consistent improvement over the baseline, with a significant improvement on occluded point tracking accuracy, showcasing GenPT’s efficiency, and the benefits of the generative casting of the task.
>
> > **A significant portion of GenPT's SOTA claim relies on the custom TAP-Vid Sliding Occluder Benchmark introduced by the authors, which is specifically designed to highlight its strength in occlusion handling. While useful, the novelty of the benchmark means the competitive results require independent verification across established, universally adopted benchmarks.**
>
> Most of the universally adopted point tracking benchmarks lack ground truth for occluded points. We include Point Odyssey and Dynamic Replica since they contain ground truth for occluded points—however, these are synthetic datasets. To substantiate our hypothesis of the real-world benefits of generative point tracking, particularly in cases of uncertainty (e.g., occluded points), we introduce the TAP-Vid Sliding Occluder benchmark, which largely contains real-world data. We are committed to releasing the benchmark for further validation.
>
> > **Q1. Have the authors explored an adaptive sampling strategy where multiple samples ('Best-of-N') are only generated in windows where the model's initial predicted uncertainty (variance/confidence) is above a certain threshold, rather than sampling N times in every window?**
>
> We have not tried this idea but it is very clever! Beyond beam search, we did not explore more exotic ways of exploiting the generative capabilities of the model because we wanted to focus on what we believe are the main basic features that generative modelling in point tracking provides: capturing multi-modality and improved occluded point tracking performance, which we show both qualitatively and quantitatively.

---

### Author Response · Authors · 2025-11-19
**Preliminary message to all reviewers**

We thank the reviewers for their time and effort in evaluating our paper. In response to some questions and concerns—such as Reviewer 837U’s concern of the lack of a comparison to CoTracker3 with random perturbations injected during test time—we have uploaded a revised manuscript that includes additional experiments in the appendix, as well as some clarifications in both the main paper and appendix.

Text highlighted in red represents modifications to existing content, and text highlighted in blue represents newly added content.

Specifically, we have added the following (highlighted in blue):
1. A comparison between GenPT and CoTracker3 with random perturbations at test time. The comparisons cover the best of N setting and a worst of N setting (Section A.14 and Tables 14-16).
    - GenPT consistently outperforms CoTracker3 + random perturbations for both best of N and worst of N, while experiencing a much lower reduction in tracking accuracy as N increases in worst of N, especially on occluded points. In some cases, there is more than a 20% percentage difference in tracking accuracy. For example, in best-of-10 (guided by oracle) GenPT has 59.2% on d_occ_avg in Dynamic Replica while CoTracker3 has 38.7%. In a single shot, GenPT and CoTracker3 have 53.8% and 31.8% on d_occ_avg in Dynamic Replica, respectively. In other words, in areas of uncertainty, GenPT is capable of capturing meaningful modes, unlike CoTracker3 with test-time random perturbations.
2. A worst of N comparison between GenPT and its discriminative variant with random perturbations at test time (Table 13 and a paragraph in A.13).
    - GenPT consistently outperforms its discriminative variant with test-time random perturbations, experiencing a much lower reduction in tracking accuracy as N increases, especially on occluded points. In some cases, there is more than a 20% percentage difference in tracking accuracy. Similar to above, this shows that GenPT is capable of capturing meaningful modes in areas of uncertainty, unlike its discriminatively-trained counterpart with test-time random perturbations.
3. Best of N comparison on a subset of the TAP-Vid sliding occluder benchmark between GenPT and its discriminative variant with random perturbations at test time (Table 11-right and Table 12-right). The table captions include additional text to reflect this addition.
    - GenPT consistently outperforms its discriminatively-trained variant with test-time random perturbations as N increases, especially on occluded points.
4. "w. test-time noise" in Table 11 and 12 for clarity.

For modifications to existing content, we have done the following (highlighted in red):
1. Provided the correct definition of the form of conditional flow matching being used in L147, as well as including references to [1, 2].
2. Corrected initially mis-reported numbers in Tables 7, 11, and 12 for the discriminative variant of GenPT. The corrected numbers are slightly higher than before, but this does not change the story.
3. "qualitative" in L1072.
4. Although not highlighted in red, we have shrunk Algorithm 1 in the appendix so that it fits better in the page.

[1] Qihao Liu, Xi Yin, Alan Yuille, Andrew Brown, and Mannat Singh. Flowing from Words to Pixels: A Noise-Free Framework for Cross-Modality Evolution. In CVPR, pp. 2755–2765, 2025.


[2] Michael S Albergo, Mark Goldstein, Nicholas M Boffi, Rajesh Ranganath, and Eric Vanden-Eijnden. Stochastic interpolants with data-dependent couplings. ICML, 2024.

---

### Meta-Review · Area_Chair_jveQ · 2026-01-09

**Summary:**

This paper addresses a meaningful and timely limitation of current discriminative point tracking methods, namely their inability to represent uncertainty and multi-modal hypotheses in ambiguous or occluded regions. Framing point tracking through a probabilistic lens is conceptually appealing, and the authors provide extensive empirical evaluations across multiple datasets, with particularly strong oracle-level performance on occluded points. The paper also introduces a modified flow-matching objective and several architectural components, which are carefully ablated.

However, despite these strengths, AC finds that the paper’s central claim of a generative reformulation of point tracking is not sufficiently substantiated, which ultimately limits the contribution.

While the method is described as generative, the actual learning and inference procedure remains closer to deterministic iterative refinement with stochastic perturbation than to a true generative process. Unlike diffusion or flow-based generative models that learn a mapping from pure noise to data and capture meaningful stochastic dynamics, GenPT starts from queries that already encode strong spatial identity. The injected noise acts primarily as a small regularizer around an already meaningful initialization, rather than as a latent variable governing diverse trajectory generation. As a result, the proposed flow-matching formulation appears functionally equivalent to supervised refinement of existing trackers (e.g., CoTracker-style models) with added Gaussian perturbation, rather than a fundamentally new modeling paradigm.

Moreover, the empirical evidence does not clearly demonstrate that the observed multi-modality is learned rather than an artifact of inference-time noise injection. The reported Best-of-N improvements could plausibly be achieved by simply running an existing tracker multiple times with random perturbations, yet no such baseline (e.g., CoTracker3 + inference-time noise) is provided. In standard single-sample evaluation—the most relevant setting for practical deployment—the gains are small and inconsistent, which further weakens the case for a learned generative advantage.

From a practical perspective, the method also raises concerns. There is a substantial and persistent gap between Oracle and Greedy performance, indicating that the model struggles to assess the quality of its own generated trajectories. Consequently, the multi-modal predictions are difficult to exploit in real-world systems. In addition, the advertised efficiency benefits hold only for single-sample inference; achieving the reported accuracy improvements requires Best-of-N sampling, which significantly increases runtime and can negate the claimed speed advantage over discriminative baselines.

Finally, the presentation and framing somewhat overstate the novelty of the contribution. The heavy use of generative modeling terminology and notation contrasts with the relatively modest algorithmic change, making the work harder to follow and potentially misleading in terms of its actual scope. While the newly introduced occlusion-focused benchmark is useful, a significant portion of the SOTA claim relies on this custom evaluation, and broader validation on established benchmarks would be needed to confirm generality.

In summary, while the paper explores an interesting direction and highlights an important limitation of existing point trackers, the generative framing is not convincingly supported by either the methodology or the empirical evidence. The approach appears best characterized as a stochastic extension of existing discriminative trackers, with limited and inconsistent practical gains. For these reasons, AC recommends Reject.

**Reviewer Concerns:**

Best of N comparison on a subset of the TAP-Vid was addressed.

**Reviewer Scores:**

The reviewer might not change the score.

---

### Decision · Program_Chairs · 2026-01-26

Reject